# Two-Timescale Networks for Nonlinear Value Function Approximation

**Wesley Chung, Somjit Nath, Ajin George Joseph and Martha White**
Department of Computing Science
University of Alberta

## Abstract

A key component for many reinforcement learning agents is to learn a value function, either for policy evaluation or control. Many of the algorithms for learning values, however, are designed for linear function approximation—with a fixed basis or fixed representation. Though there have been a few sound extensions to nonlinear function approximation, such as nonlinear gradient temporal difference learning, these methods have largely not been adopted, eschewed in favour of simpler but not sound methods like temporal difference learning and Q-learning. In this work, we provide a two-timescale network (TTN) architecture that enables linear methods to be used to learn values, with a nonlinear representation learned at a slower timescale. The approach facilitates the use of algorithms developed for the linear setting, such as data-efficient least-squares methods, eligibility traces and the myriad of recently developed linear policy evaluation algorithms, to provide nonlinear value estimates. We prove convergence for TTNs, with particular care given to ensure convergence of the fast linear component under potentially dependent features provided by the learned representation. We empirically demonstrate the benefits of TTNs, compared to other nonlinear value function approximation algorithms, both for policy evaluation and control.

## 1 Introduction

Value function approximation—estimating the expected returns from states for a policy—is heavily reliant on the quality of the representation of state. One strategy has been to design a basis—such as radial basis functions (Sutton and Barto, 1998) or a Fourier basis (Konidaris et al., 2011)—for use with a linear function approximator and temporal difference (TD) learning (Sutton, 1988). For low-dimensional observation vectors, this approach has been effective, but can be onerous to extend to high-dimensional observations, potentially requiring significant domain expertise. Another strategy has been to learn the representation, such as with basis adaptation or neural networks. Though there is still the need to specify the parametric form, learning these representations alleviates the burden of expert specification. Further, it is more feasible to scale to high-dimensional observations, such as images, with neural networks (Mnih et al., 2015; Silver et al., 2016). Learning representations necessitates algorithms for nonlinear function approximation.

Despite the deficiencies in specification for fixed bases, linear function approximation for estimating value functions has several benefits over nonlinear estimators. They enable least-squares methods, which can be much more data-efficient for policy evaluation (Bradtke and Barto, 1996; Szepesvari, 2010; van Seijen and Sutton, 2015), as well as robust to meta-parameters (Pan et al., 2017). Linear algorithms can also make use of eligibility traces, which can significantly speed learning (Sutton, 1988; Dann et al., 2014; White and White, 2016), but have not been able to be extended to nonlinear value function approximation. Additionally, there have been a variety of algorithms derived for the linear setting, both for on-policy and off-policy learning (Sutton et al., 2009; Maei, 2011; van Seijen and Sutton, 2014; van Hasselt et al., 2014; Mahadevan et al., 2014; Sutton et al., 2016; Mahmood et al., 2017). These linear methods have also been well-explored theoretically (Tsitsiklis and Van Roy, 1997; Maei, 2011; Mahmood and Sutton, 2015; Yu, 2015) and empirically (Dann et al., 2014; White and White, 2016), with some insights into improvements from gradient methods (Sutton et al., 2009), true-online traces (van Seijen and Sutton, 2014) and emphatic weightings (Sutton et al., 2016). These algorithms are easy to implement, with relatively simple objectives. Objectives for nonlinear value function approximation, on the other hand, can be quite complex (Maei et al., 2009), resulting in

more complex algorithms (Menache et al., 2005; Di Castro and Mannor, 2010; Bhatnagar et al., 2013) or requiring a primal-dual formulation as has been done for control (Dai et al., 2017).

In this work, we pursue a simple strategy to take advantage of the benefits of linear methods, while still learning the representation. The main idea is to run two learning processes in parallel: the first learns nonlinear features using a surrogate loss and the second estimates the value function as a linear function of those features. We show that these Two-timescale Networks (TTNs) converge, because the features change on a sufficiently slow scale, so that they are effectively fixed for the fast linear value function estimator.

Similar ideas have previously been explored for basis adaptation, but without this key aspect of TTNs—namely the separation of the loss for the representation and value function. This separation is critical because it enables simpler objectives—for which the gradient can be easily sampled—to drive the representation, but still enables use of the mean squared projected Bellman error (MSPBE)—on which all the above linear algorithms are based. This separation avoids the complexity of the nonlinear MSPBE, but maintains the useful properties of the (linear) MSPBE. A variety of basis adaptation approaches have used a two-timescale approach, but with the same objective for the representation and the values (Menache et al., 2005; Di Castro and Mannor, 2010; Bhatnagar et al., 2013; J et al., 2016). Yu and Bertsekas (2009) provided algorithms for basis adaptation using other losses, such as Bellman error using Monte carlo samples, taking derivatives through fixed point solutions for the value function. Levine et al. (2017) periodically compute a closed form least-squares solution for the last layer of neural network, with a Bayesian update to prevent too much change. Because these methods did not separate the value learn and basis adaptation, the resulting algorithms are more complex. The strategy of using two different heads—one to drive the representation and one to learn the values—has yet to be systematically explored.

We show that TTNs are a promising direction for nonlinear function approximation, allowing us to leverage linear algorithms while retaining the flexibility of nonlinear function approximators. We first discuss a variety of possible surrogate losses, and their potential for learning a useful representation. We then show that TTNs converge, despite the fact that a linear algorithm is used with a changing representation. This proof is similar to previous convergence proofs for policy evaluation, but with a relaxation on the requirement that features be independent, which is unlikely for learned features. We then show empirically that TTNs are effective compared to other nonlinear value function approximations and that they can exploit several benefits of linear value approximations algorithms. In particular, for both low-dimensional and high-dimensional (image-based) observations, we show (a) the utility of least-squares (or batch) methods, (b) advantages from eligibility traces and (c) gains from being able to select amongst different linear policy evaluation algorithms. We demonstrate that TTNs can be effective for control with neural networks, enabling use of fitted Q-iteration within TTNs as an alternative to target networks.

## 2 BACKGROUND

We assume the agents act in a finite Markov Decision Process (MDP), with notation from (White, 2017). The dynamics of the MDP are defined by the 3-tuple $(\mathcal{S}, \mathcal{A}, P)$, where $\mathcal{S}$ is the set of states, $\mathcal{A}$ the set of actions and $P : \mathcal{S} \times \mathcal{A} \times \mathcal{S} \mapsto [0, 1]$ the transition probability function. The task in this environment is defined by a reward function $R : \mathcal{S} \times \mathcal{A} \times \mathcal{S} \mapsto \mathbb{R}$ and a discount function $\gamma : \mathcal{S} \times \mathcal{A} \times \mathcal{S} \mapsto [0, 1]$. At each time step, the agent takes an action $A_t$ according to a policy $\pi : \mathcal{S} \times \mathcal{A} \mapsto [0, 1]$ and the environment returns reward $R_{t+1}$, next state $S_{t+1}$ and discount $\gamma_{t+1}$.

The goal in policy evaluation is to compute the value function: the expected sum of discounted rewards from every state under a fixed policy $\pi$. The value function $V_\pi : \mathcal{S} \to \mathbb{R}$ is defined recursively from each state $s \in \mathcal{S}$ as

$$V_\pi(s) \stackrel{\text{def}}{=} \mathbb{E}[R_{t+1} + \gamma_{t+1} V_\pi(S_{t+1})|S_t = s] = \sum_{a \in \mathcal{A}} \pi(s, a) \sum_{s' \in \mathcal{S}} P(s, a, s')(r + \gamma V_\pi(s')). \quad (1)$$

When using linear function approximation, this goal translates into finding parameters $\mathbf{w} \in \mathbb{R}^d$ to approximate the value function

$$\hat{V}(s) \stackrel{\text{def}}{=} \mathbf{x}(s)^\top \mathbf{w} \approx V_\pi(s) \qquad \text{where } \mathbf{x} : \mathcal{S} \to \mathbb{R}^d \text{ is a feature function.} \quad (2)$$

More generally, a nonlinear function $\hat{V}(s)$ could be learned to estimate $V_\pi$.

To formulate this learning problem, we need to consider the objective for learning the function $\hat{V}$. Let $\boldsymbol{V_\pi}, \hat{V} \in \mathbb{R}^{|\mathcal{S}|}$ be the vectors of values for $V_\pi, \hat{V}$. The recursive formula (1) defines a Bellman operator $\mathbf{B}_\pi$ where the fixed point satisfies $\mathbf{B}_\pi \boldsymbol{V_\pi} = \boldsymbol{V_\pi}$. Consider a restricted value function class, such as the set of linear value functions $\hat{V} \in \mathcal{F} = \{\mathbf{X}\mathbf{w} \mid \mathbf{w} \in \mathbb{R}^d\}$ where $\mathbf{X} \in \mathbb{R}^{|\mathcal{S}| \times d}$ is a matrix with the $i$-th row set to $\mathbf{x}(s)$ for $i$th state $s \in \mathcal{S}$. Then, it may no longer be possible to satisfy the recursion. Instead, an alternative is to find a projected fixed point $\mathbf{\Pi}_\mathcal{F} \mathbf{B}_\pi \hat{V} = \hat{V}$ where the projection operator $\mathbf{\Pi}_\mathcal{F}$ projects $\mathbf{B}_\pi \hat{V}$ to the space spanned by this linear basis:

$$\mathbf{\Pi}_\mathcal{F} \boldsymbol{V} \overset{\text{def}}{=} \underset{\bar{\boldsymbol{V}} \in \mathcal{F}}{\arg\min} \|\bar{\boldsymbol{V}} - \boldsymbol{V}\|_d^2 \tag{3}$$

where $\mathbf{d} \in \mathbb{R}^{|\mathcal{S}|}$ is a vector which weights each state in the weighted norm $\|\boldsymbol{V}\|_d^2 = \sum_{s \in \mathcal{S}} \mathbf{d}(s)\boldsymbol{V}(s)^2$. Many linear policy evaluation algorithms estimate this projected fixed point, including TD (Sutton, 1988), least-squares TD (Bradtke and Barto, 1996) and gradient TD (Sutton et al., 2009).

The objective formulated for this projected fixed-point, however, is more complex for nonlinear function approximation. For linear function approximation, the projection operator simplifies into a closed form solution involving only the features $\mathbf{X}$. Letting $\delta_t = R_{t+1} + \gamma \hat{V}(S_{t+1}) - \hat{V}(S_t)$, the resulting mean-squared projected Bellman error (MSPBE) can be written as

$$\text{MSPBE}(\mathbf{w}) \overset{\text{def}}{=} \|\mathbf{\Pi}_\mathcal{F} \mathbf{B}_\pi \hat{V} - \hat{V}\|_d^2 = \mathbb{E}[\delta_t \mathbf{x}_t] \, \mathbb{E}[\mathbf{x}_t \mathbf{x}_t^\top]^{-1} \, \mathbb{E}[\delta_t \mathbf{x}_t] \tag{4}$$

where $\mathbb{E}[\delta_t \mathbf{x}_t] = \sum_{s \in \mathcal{S}} \mathbf{d}(s) \, \mathbb{E}[\delta_t | S_t = s] \mathbf{x}(s)$. For nonlinear function classes, the projection does not have a closed form solution and may be expensive to compute. Further, the projection involves the value function parameters, so the projection changes as parameters change. The nonlinear MSPBE and resulting algorithm are more complex (Maei et al., 2009), and have not seen widespread use.

Another option is simply to consider different objectives. However, as we discuss below, other objectives for learning the value function either are similarly difficult to optimize or provide poor value estimates. In the next section, we discuss some of these alternatives and introduce Two-timescale Networks as a different strategy to enable nonlinear value function approximation.

## 3 TWO-TIMESCALE NETWORKS AND SURROGATE OBJECTIVES

We first introduce Two-timescale Networks (TTNs), and then describe different surrogate objectives that can be used in TTNs. We discuss why these surrogate objectives within TTNs are useful to drive the representation, but are not good replacements for the MSPBE for learning the value function.

TTNs use two concurrent optimization processes: one for the parameters of the network $\boldsymbol{\theta}$ and one for the parameters of the value function $\mathbf{w}$. The value function is approximated as $\hat{V}(s) \overset{\text{def}}{=} \mathbf{x}_{\boldsymbol{\theta}}(s)^\top \mathbf{w}$ where the features $\mathbf{x}_\theta : \mathcal{S} \to \mathbb{R}^d$ are a parametrized function and $\boldsymbol{\theta} \in \mathbb{R}^m$ is adjusted to provide better features. For a neural network, $\boldsymbol{\theta}$ consists of all the parameters in the hidden layers, to produce the final hidden layer $\mathbf{x}_{\boldsymbol{\theta}}(s)$. The two optimization processes maintain different time scales, with the parameters $\boldsymbol{\theta}$ for the representation changed as a *slow* process, and the parameters $\mathbf{w}$ for the value estimate changed as a *fast* process relative to $\boldsymbol{\theta}$.

The separation between these two processes could be problematic, since the target problem—estimating the value function—is not influencing the representation! The slow process is driven by a completely separate objective than the fast process. However, the key is to select this surrogate loss for the slow process so that it is related to the value estimation process, but still straightforward to compute the gradient of the loss. We use $\hat{V}(s)$ as the output of the fast part, which corresponds to the value estimate used by the agent. To distinguish, $\hat{Y}(s)$ denotes the output for the slow-part (depicted in Figure 1), which may or may not be an estimate of the value, as we discuss below.

Consider first the mean-squared TD error (MSTDE), which corresponds to $\sum_{s \in \mathcal{S}} \mathbf{d}(s) \, \mathbb{E}[\delta_t^2 | S_t = s]$. Notice that this does not correspond to the mean-squared Bellman error (MSBE), for which it is more difficult to compute gradients $\|\mathbf{B}_\pi \hat{V} - \hat{V}\|_d^2 = \sum_{s \in \mathcal{S}} \mathbf{d}(s) \, (\mathbb{E}[\delta_t | S_t = s])^2$. Using the MSTDE as a surrogate loss, with $\hat{Y}(s) = \mathbf{x}_{\boldsymbol{\theta}}(s)^\top \bar{\mathbf{w}}$, the slow part of the network minimizes

$$L_{\text{slow}}(\boldsymbol{\theta}) = \min_{\bar{\mathbf{w}} \in \mathbb{R}^d} \sum_{s \in \mathcal{S}} \mathbf{d}(s) \, \mathbb{E}[\delta_t(\boldsymbol{\theta}, \bar{\mathbf{w}})^2 | S_t = s] \quad \triangleright \ \delta_t(\boldsymbol{\theta}, \bar{\mathbf{w}}) \overset{\text{def}}{=} R_{t+1} + \gamma_{t+1} \mathbf{x}_{\boldsymbol{\theta}}(S_{t+1})^\top \bar{\mathbf{w}} - \mathbf{x}_{\boldsymbol{\theta}}(S_t)^\top \bar{\mathbf{w}}.$$

This slow part has its own weights $\bar{\mathbf{w}}$ associated with estimating the value function, but learned instead according to the MSTDE. The advantage here is that stochastic gradient descent on the MSTDE is straightforward, with gradient $\delta_t \nabla_{\{\boldsymbol{\theta}, \bar{\mathbf{w}}\}} [\gamma_{t+1} \hat{Y}(S_{t+1}) - \hat{Y}(S_t)]$ where $\nabla_{\{\boldsymbol{\theta}, \bar{\mathbf{w}}\}} \hat{Y}(S_t)$ is the gradient of the neural network, including the head of the slow part which uses weights $\bar{\mathbf{w}}$. Using the MSTDE has been found to provide worse value estimates than the MSPBE—which we re-affirm in our experiments. It could, nonetheless, play a useful role as a surrogate loss, where it can inform the representation towards estimating values.

There are a variety of other surrogate losses that could be considered, related to the value function. However, many of these losses are problematic to sample incrementally, without storing large amounts of data. For example, the mean-squared return error (MSRE) could be used, which takes samples of return and minimizes mean-squared error to those sampled returns. Obtaining such returns requires waiting many steps, and so delays updating the representation for the current state. Another alternative is the MSBE. The gradient of the nonlinear MSBE

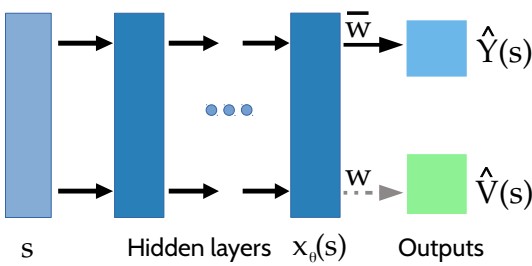

Figure 1: Two-Timescale Network architecture

is not as complex as the gradient of the nonlinear MSPBE, because it does not involve the gradient of a projection. However, it suffers from the double sampling problem: sampling the gradient requires two independent samples. For these reasons, we explore the MSTDE as the simplest surrogate loss involving the value function.

Finally, surrogate losses could also be defined that are not directly related to the value function. Two natural choices are losses based on predicting the next state and reward. The output of the slow part could correspond to a vector of values, such as $Y_t = S_{t+1} \in \mathbb{R}^n$ or $Y_t = \begin{bmatrix} S_{t+1} \\ R_{t+1} \end{bmatrix}$. The ability to predict the next state and reward is intuitively useful for enabling prediction of value, that also has some theoretical grounding. Szepesvari (2010, Section 3.2.1) shows that the Bellman error is small, if the features can capture a horizon of immediate rewards and expected next states. For linear encoders, Song et al. (2016) show that an optimal set of features enables predictions of next state and reward.

More generally, learning representations using auxiliary tasks or self-supervised tasks have had some successes in RL, such as using pixel control (Jaderberg et al., 2016) or classifying the temporal distance between frames (Aytar et al., 2018). In computer vision, Gidaris et al. (2018) showed that using rotated images as self-supervised tasks produced a useful representation for the main loss, without training the representation with the main loss. Any of these self-supervised tasks could also be used for the surrogate objective, and motivate that separating out representation learning does not degrade performance. For now, we restrict focus on simpler surrogate objectives, as the main purpose of this work is to demonstrate that the separation in TTNs is a sound approach for learning values.

## 4 Convergence of Two-Timescale Network Algorithm

Training TTNs is fully online, using a single transition from the environment at a time. Projected stochastic gradient descent is used to reduce the surrogate loss, $L_{\text{slow}}(\boldsymbol{\theta})$ and a linear policy evaluation algorithm, such as GTD2 or TD($\lambda$), is coupled to the network where the prediction vector $\mathbf{w}$ is calibrated proportional to $-\nabla_{\mathbf{w}} \text{MSPBE}_{\boldsymbol{\theta}}(\mathbf{w})$. The full procedure is summarized in Algorithm 1, in Appendix A. Regarding the convergence of TTNs, a few remarks are in order:

1. The network needs to evolve sufficiently slowly relative to the linear prediction weights. In our theoretical analysis, this is achieved by ensuring that the step sizes $\xi_t$ and $\alpha_t$ of the network and the linear policy evaluation algorithm respectively decay to zero at different rates. In particular, $\xi_t / \alpha_t \to 0$ as $t \to \infty$. With this relative disparity in magnitudes, one can assume that the network is essentially quasi-static, while the faster linear component is equilibrated relative to the static features.

2. The linear prediction algorithms need to converge for any set of features provided by the neural network, particularly linearly dependent features. This induces a technical bottleneck since linear independence of the features are a necessary condition for the convergence of the prediction methods GTD and GTD2. We overcome this by following a differential inclusion based analysis for GTD2.

3. Finally, we need to guarantee the stability of the iterates (both feature vector $\theta_t$ and the prediction vector $\mathbf{w}_t$) and this is ensured by projecting the iterates to respective compact, convex sets.

The analysis for the convergence of the neural network is general, enabling any network architectures that are twice continuously differentiable. We prove that the TTNs converge asymptotically to the stable equilibria of a projected ODE which completely captures the mean dynamics of the algorithm.

We now state our main result (for notations and technical details, please refer Appendix B). The results are provided for cases when TD($\lambda$) or GTD2 is used as the linear prediction method. However, note that similar results can be obtained for other linear prediction methods.

**Theorem 1.** *Let $\bar{\theta} = (\theta, \bar{\mathbf{w}})^\top$ and $\Theta \subset \mathbb{R}^{m+d}$ be a compact, convex subset with smooth boundary. Let the projection operator $\Gamma^\Theta$ be Frechet differentiable and $\widehat{\Gamma}_{\bar{\theta}}^\Theta(-\frac{1}{2}\nabla L_{slow})(\bar{\theta})$ be Lipschitz continuous. Also, let Assumptions 1-3 hold. Let $\mathcal{K}$ be the set of asymptotically stable equilibria of the following ODE contained inside $\Theta$:*

$$\frac{d}{dt}\bar{\theta}(t) = \widehat{\Gamma}_{\bar{\theta}(t)}^\Theta(-\frac{1}{2}\nabla_{\bar{\theta}}L_{slow})(\bar{\theta}(t)), \quad \bar{\theta}(0) \in \overset{\circ}{\Theta} \text{ and } t \in \mathbb{R}_+.$$

*Then the stochastic sequence $\{\bar{\theta}_t\}_{t\in\mathbb{N}}$ generated by the TTN converges almost surely to $\mathcal{K}$ (sample path dependent). Further,*

**TD($\lambda$) Convergence:** *Under the additional Assumption 4-TD($\lambda$), we obtain the following result: For any $\lambda \in [0, 1]$, the stochastic sequence $\{\mathbf{w}_t\}_{t\in\mathbb{N}}$ generated by the TD($\lambda$) algorithm (Algorithm 2) within the TTN setting converges almost surely to the limit $\mathbf{w}^*$, where $\mathbf{w}^*$ satisfies*

$$\Pi_{\bar{\theta}^*}T^{(\lambda)}(\Phi_{\bar{\theta}^*}\mathbf{w}^*) = \Phi_{\bar{\theta}^*}\mathbf{w}^*, \tag{5}$$

*with $\bar{\theta}^* \in \mathcal{K}$ (sample path dependent).*

## 5 EXPERIMENTS

We investigate the performance of TTNs versus a variety of other nonlinear policy evaluation algorithms, as well as the impact of choices within TTNs. We particularly aim to answer (a) is it beneficial to optimize the MSPBE to obtain value estimates, rather than using value estimates from surrogate losses like the MSTDE; (b) do TTNs provide gains over other nonlinear policy evaluation algorithms; and (c) can TTNs benefit from the variety of options in linear algorithms, including least-squares approaches, eligibility traces and different policy evaluation algorithms. More speculatively, we also investigate if TTNs can provide a competitive alternative to deep Q-learning in control.

Experiments were performed on-policy in five environments. We use three classic continuous-state domains: Puddle World, a continuous-state grid world with high-magnitude negative rewards for walking through a puddle; Acrobot, where a robot has to swing itself up; and Cartpole, which involves balancing a pole. We also use two game domains: Catcher, which involves catching falling apples; and Puck World, in which the agent has to chase a puck (Tasfi, 2016). Catcher includes both a variant with 4-dimensional observations—position and velocity of the paddle, and (x,y) of the apple— and one with image-based observations—with two consecutive 64-by-64 grayscale images as input. This domain enables us to analyze the benefit of the algorithms, on the same domain, both with low-dimensional and high-dimensional observations. We describe the policies evaluated for these domains in Appendix D. We include a subset of results in the main body, with additional results in the appendix. Results in Cartpole are similar to Acrobot; Cartpole results are only in the appendix.

The value estimates are evaluated using root-mean-squared value error (RMSVE), where value error is $(V_\pi(s) - \hat{V}(s))^2$. The optimal values for a set of 500 states are obtained using extensive rollouts from each state and the RMSVE is computed across these 500 states. For the algorithms, we use the following settings, unless specified otherwise. For the slow part (features), we minimize the mean-squared TD error (MSTDE) using the AMSGrad optimizer (Reddi et al., 2018) with $\beta_1 = 0$ and $\beta_2 = 0.99$. The network weights use Xavier initialization (Glorot and Bengio, 2010); the weights for the fast part were initialized to 0. In Puddle World, the neural network consists of a single hidden layer of 128 units with ReLU activations. In the other environments, we use 256 units instead. To choose hyperparameters, we first did a preliminary sweep on a broad range and then chose a smaller range where the algorithms usually made progress, summarized in Appendix D. Results are reported for hyperparameters in the refined range, chosen based on RMSVE over the latter half of a run with shaded regions corresponding to one standard error.

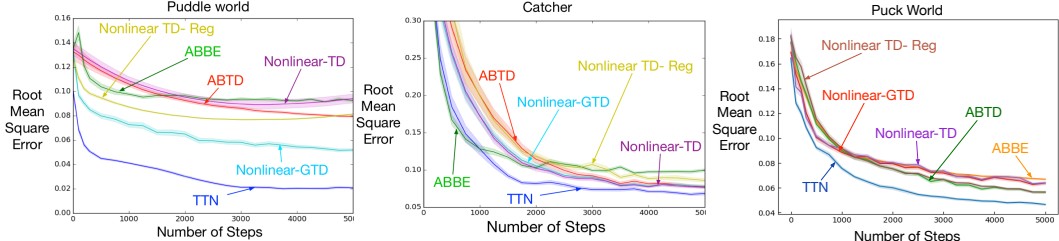

Figure 2: TTN comparison to other nonlinear value function approximation algorithms. For the TTN, MSTDE is used as the surrogate loss for the slow part of the network (feature learning) and LSTD is used for the fast part.

**TTN vs. competitors.**   We compare to the following algorithms: nonlinear TD, nonlinear GTD (Maei et al., 2009), Adaptive Bases (ABBE and ABTD) (Di Castro and Mannor, 2010), nonlinear TD + LSTD regularization (inspired by Levine et al. (2017)). We describe these algorithms in more depth in Appendix D. All of the algorithms involve more complex updates compared to TTNs, except for nonlinear TD, which corresponds to a semi-gradient TD update with nonlinear function approximation. For TTNs, we use LSTD for the linear, fast part.

In Figure 2, TTN is able to perform as well or better than the competitor algorithms. Especially in Puddle World, its error is significantly lower than the second best algorithm. Interestingly, Nonlinear GTD also performs well across domains, suggesting an advantage for theoretically-sound algorithms.

**The utility of optimizing the MSPBE.**   First, we show that the TTN benefits from having a second head learning at a faster timescale. To do so, we compare the prediction errors of using TTN, with the fast process optimizing the MSPBE (using LSTD) and the slow one optimizing the MSTDE, and one trained end-to-end using the MSTDE with AMSGrad. As a baseline, we include TTN with a fixed representation (a randomly initialized neural network) to highlight that the slow process is indeed improving the representation. We also include results for optimizing the MSTDE with the fixed representation.

In Figure 3, we see that optimizing the MSPBE indeed gives better results than optimizing the MSTDE. Additionally, we can conclude that using the MSTDE, despite being a poor objective to learn the value function, can still be effective for driving feature-learning since it outperforms the fixed representation.

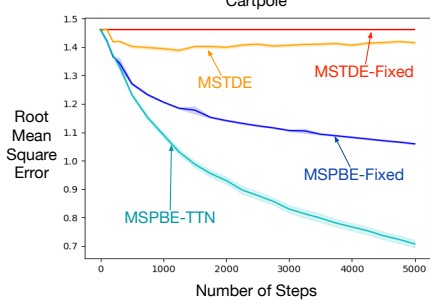

Figure 3: Comparison of MSPBE and MSTDE.

**Linear algorithms and eligibility traces.** TTNs give us the flexibility to choose any linear policy evaluation algorithm for the fast part. We compare several choices: TD, least-squares TD (LSTD) (Bradtke and Barto, 1996), forgetful LSTD (FLSTD) (van Seijen and Sutton, 2015), emphatic TD (Sutton et al., 2016), gradient TD (the TDC variant) (Sutton et al., 2009) and their true-online versions (van Seijen and Sutton, 2014; van Hasselt et al., 2014) to learn the value function. GTD and ETD are newer temporal difference methods which have better convergence properties and can offer increased stability. The true-online variants modify the update rules to improve the behavior of the algorithms when learning online and seem to outperform their counterparts empirically (van Seijen and Sutton, 2014). Least-squares methods summarize past interaction, but are often avoided due to quadratic computation in the number of features. For TTNs, however, there is no computational disadvantage to using LSTD methods, for two reasons. It is common to choose deep but skinny architectures (Mnih et al., 2015; Hessel et al., 2017). Furthermore, if the last layer is fully connected, then we already need to store $O(d^2)$ weights and use $O(d^2)$ time to compute a forward pass—the same as LSTD. We include FLSTD, which progressively forgets older interaction, as this could be advantageous when the feature representation changes over time. For TTN, incremental versions of the least-squares algorithms are used to maintain estimates of the required quantities online (see appendix D).

All of these linear algorithms can use eligibility traces to increase their sample efficiency by propagating TD errors back in time. The trace parameter $\lambda$ can also provide a bias-variance tradeoff for the value estimates (Sutton, 1988; Dann et al., 2014). For nonlinear function approximation, eligibility traces can no longer be derived for TD. Though invalid, we can naively extend them to this case by keeping one trace per weight, giving us nonlinear TD($\lambda$).

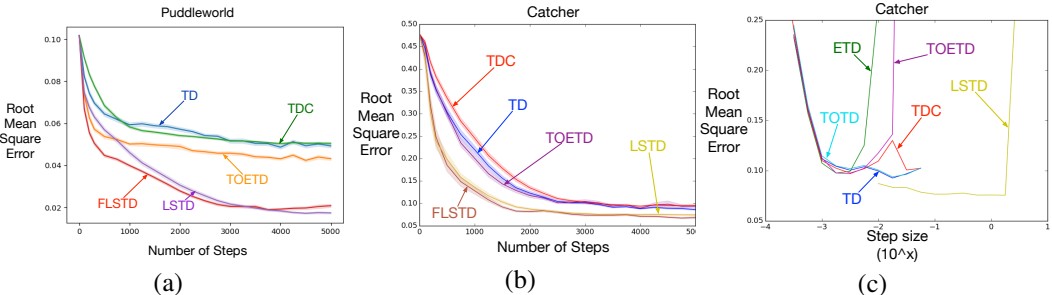

Figure 4: a) & b) Linear methods on Puddle World and Catcher. c) Step size sensitivity in Catcher.

The results overall indicate that TTNs can benefit from the ability to use different linear policy evaluation algorithms and traces, in particular from the use of least-squares methods as shown in Figure 4 for Puddle World and Catcher. The dominance of LSTD versus the other linear algorithms is consistent, including in terms of parameter sensitivity, persists for the other three domains. We additionally investigated sensitivity to $\lambda$, and found that most of the TTN variants benefit from a nonzero $\lambda$ value and, in many cases, the best setting is high, near 1. One exception is the least-squares methods, where LSTD performs similarly for most values of $\lambda$. Nonlinear TD($\lambda$), on the other hand, performs markedly worse as $\lambda$ increases. This is unsurprising considering the naive addition of eligibility traces is unsound. We include these sensitivity plots in the appendix, in Figure **??**.

**Surrogate loss functions.** For all the previous experiments, we optimized the MSTDE for the slow part of the network, but as discussed in Section 3, other objectives can be used. We compare a variety of objectives, by choosing different $Y_t$, including $Y_t = R_{t+1}$ (Reward); $Y_t = S_{t+1}$ (Next State); and $Y_t = R_{t+1} + \hat{Y}(S_{t+1})$. (Semi-gradient MSTDE). In Puck World, in Figure 5 a), we can see that every auxiliary loss performed well. This does not appear to be universally true, as in Acrobot we found that the MSTDE was a less effective surrogate loss, leading to slower learning (see Figure 5 b). Alternate losses such as the semi-gradient MSTDE and next state predictions were more successful in that domain. These results suggest that there is no universally superior surrogate loss and that choosing the appropriate one can yield benefits in certain domains.

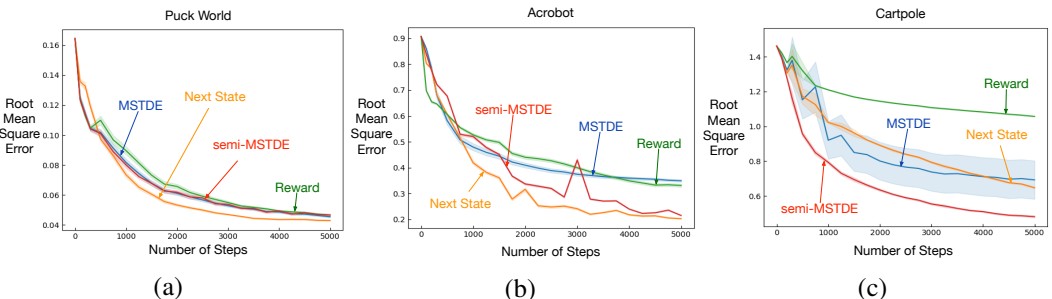

Figure 5: a), b) & c) Comparison of surrogate losses on Puck World, Acrobot and Cartpole

**Control** Although the focus of this work is policy evaluation, we also provide some preliminary results for the control setting. For control, we include some standard additions to competitor learning algorithms to enable learning with neural networks. The DQN algorithm (Mnih et al., 2015) utilizes two main tricks to stabilize training: experience replay—storing past transitions and replaying them multiple times—and a target network—which keeps the value function in the Q-learning targets fixed, updating the target network infrequently (e.g., every $k = 10,000$ steps).

We use an alternative strategy to target networks for TTN. The use of a target network is motivated by fitted Q-iteration (FQI) (Ernst et al., 2005), which updates towards fixed Q-values with one sweep through a batch of data. TTNs provide a straightforward mechanism to instead directly use FQI, where we can solve for the weights on the entire replay buffer, taking advantage of the closed form solution for linear regression towards the Q-values from the last update. Batch FQI requires storing all data, whereas we instead have a sliding window of experience. We therefore additionally incorporate a regularization term, which prevents the weights from changing too significantly between updates, similarly to Levine et al. (2017). Each FQI iteration requires solving a least squares problem on the entire buffer, an operation costing $O(nd^2)$ computation where $d$ is the number of features in the last layer of the network and $n$ is the size of the buffer; we update the network every $k$ steps, which reduces the per-step computation to $O(nd^2/k)$. The slow part drives feature-learning by minimizing the semi-gradient MSTDE for state-action values. As another competitor, we include

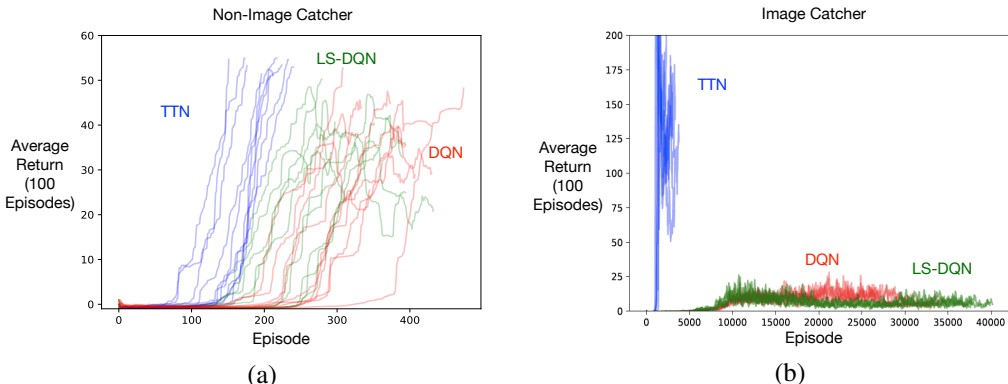

Figure 6: a) Comparison of returns obtained by each algorithm on a) non-image Catcher and b) image Catcher.

LS-DQN (Levine et al., 2017), a DQN variant which also utilizes adjustments to the final layer's weights towards the FQI solution, similar to TTN-FQI.

The experimental details differ for control. On nonimage Catcher, we do a sweep over $\alpha_{slow}$ and $\lambda_{reg}$, the regularization parameter, for TTN and sweep over the learning rate and the number of steps over which $\epsilon$ is annealed for DQN. On image Catcher, runs require significantly more computation so we only tune hyperparameters by hand. The FQI update in TTNs was done every 1000 (10000) steps for non-image (image) Catcher. We run each algorithm 10 times (5 times) for 200 thousand steps (10 million steps) on the non-image (image) Catcher.

We see that TTN is able to perform well on both versions of Catcher in Figure 6, particularly learning more quickly than the DQN variants. This difference is especially pronounced in the image version of catcher, where TTN is also able to achieve much higher average returns than DQN. Both algorithms seem to suffer from catastrophic forgetting later during training as the performance dips down after an initial rise, although TTN still stabilizes on a better policy. Overall, these results suggest that TTNs are a promising direction for improving sample efficiency in control, whilst still maintaining stability when training neural networks.

## 6 DISCUSSION AND CONCLUSION

In this work, we proposed Two-timescale Networks as a new strategy for policy evaluation with nonlinear function approximation. As opposed to many other algorithms derived for nonlinear value function approximation, TTNs are intentionally designed to be simple to promote ease-of-use. The algorithm combines a slow learning process for adapting features and a fast process for learning a linear value function, both of which are straightforward to train. By leveraging these two timescales, we are able to prove convergence guarantees for a broad class of choices for both the fast and slow learning components. We highlighted several cases where the decoupled architecture in TTNs can improve learning, particularly enabling the use of linear methods—which facilitates use of least-squares methods and eligibility traces.

This work has only begun the investigation into which combinations for surrogate losses and linear value function approximation algorithms are most effective. We provided some evidence that, when using stochastic approximation algorithms rather than least-squares algorithms, the addition of traces can have a significant effect within TTNs. This contrasts nonlinear TD, where traces were not effective. The ability to use traces is potentially one of the most exciting outcomes for TTNs, since traces have been so effective for linear methods. More generally, TTNs provide the opportunity to investigate the utility of the many linear value function algorithms, in more complex domains with learned representations. For example, emphatic algorithms have improved asymptotic properties (Sutton et al., 2016), but to the best of our knowledge, have not been used with neural networks.

Another promising direction for TTNs is for off-policy learning, where many value functions are learned in parallel. Off-policy learning can suffer from variance due to large magnitude corrections (importance sampling ratios). With a large collection of value functions, it is more likely that some of them will cause large updates, potentially destabilizing learning in the network if trained in an end-to-end fashion. TTNs would not suffer from this problem, because a different objective can be used to drive learning in the network. We provide some preliminary experiments in the appendix supporting this hypothesis (Appendix C.7).

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

## A  TTN Algorithm

**Algorithm 1** Training of TTNs

1: **procedure** TRAIN($\mathbf{w}, \boldsymbol{\theta}, \bar{\mathbf{w}}, \pi$)                                              ▷ $\pi$ is a fixed policy
2:     Initialize $\boldsymbol{\theta}, \bar{\mathbf{w}}$ with Xavier initialization, $\mathbf{w}$ to 0 and thestarting state $s$ according to the
    environment
3:     **while** *training* **do**
4:         $a \leftarrow$ action chosen by $\pi(s)$
5:         $r, s' \leftarrow$ Environment$(s, a)$                                              ▷ Get reward and next state
6:         $\boldsymbol{\theta}, \bar{\mathbf{w}} \leftarrow$ GradientDescent on $\mathcal{L}_{slow}$ using sample $(s, r, s')$
7:         $\mathbf{w} \leftarrow$ Update on $\mathcal{L}_{value}$ using sample $(s, r, s')$
8:         $s \leftarrow s'$
9:     **end while**
10:     **return** learned parameters $\mathbf{w}, \boldsymbol{\theta}, \bar{\mathbf{w}}$
11: **end procedure**

## B  Convergence proof of two-timescale networks

### B.1  Definitions & Notations

- Let $\mathbb{R}_+$ denote the set of non-negative real numbers, $\mathbb{N} = \{0, 1, 2, \dots\}$ and $\|\cdot\|$ denote the Euclidean norm or any equivalent norm.

- A map $f : \mathbb{R}^d \to \mathbb{R}^d$ is Lipschitz continuous if $\|f(\mathbf{x}) - f(\mathbf{y})\| \leq L(\|\mathbf{x} - \mathbf{y}\|)$, for some $L \in (0, \infty), \forall \mathbf{x}, \mathbf{y} \in \mathbb{R}^d$.

- A set-valued map $h : \mathbb{R}^d \to \{\text{subsets of } \mathbb{R}^d\}$ is called a Marchaud map, if it satisfies the following conditions:

  1. For each $\mathbf{x} \in \mathbb{R}^d$, $h(\mathbf{x})$ is convex and compact.
  2. For each $\mathbf{x} \in \mathbb{R}^d$, $\exists K \in (0, \infty)$ such that $\sup_{\mathbf{y} \in h(\mathbf{x})} \|\mathbf{y}\| \leq K(1 + \|\mathbf{x}\|)$.
  3. $h$ is upper-semicontinuous, *i.e.*, if $\{\mathbf{x}_n\}_{n \in \mathbb{N}} \to \mathbf{x}$ and $\{\mathbf{y}_n\}_{n \in \mathbb{N}} \to \mathbf{y}$, where $\mathbf{x}_n \in \mathbb{R}^d$, $\mathbf{y}_n \in h(\mathbf{x}_n), \forall n \in \mathbb{N}$, then $\mathbf{y} \in h(\mathbf{x})$.

- For $\mathbf{x}_1, \mathbf{x}_2 \in \mathbb{R}^d$ and $D \in \mathbb{R}^{k \times k}$ a diagonal matrix, we define the inner-product $< \mathbf{x}_1, \mathbf{x}_2 >_D \triangleq \mathbf{x}_1^\top D \mathbf{x}_2$. We also define the semi-norm $\|\mathbf{x}\|_D \triangleq < \mathbf{x}, \mathbf{x} >_D^{\frac{1}{2}}$. If all the diagonal elements of D are strictly positive, then $\|\cdot\|_D$ is a norm.

- For any set $X$, let $\mathring{X}$ denote the interior of $X$ and $\partial X$ denote the boundary of $X$.

- For brevity, let $\bar{\theta} = (\theta, \bar{\mathbf{w}})^\top$ and $\Phi_{\bar{\theta}}$ be the feature matrix corresponding to the feature parameter $\bar{\theta}$, *i.e.*

$$\Phi_{\bar{\theta}} \triangleq \begin{pmatrix} \mathbf{x}_\theta(s_1)^\top \\ \mathbf{x}_\theta(s_2)^\top \\ \vdots \\ \mathbf{x}_\theta(s_{|\mathcal{S}|})^\top \end{pmatrix}_{|\mathcal{S}| \times d}, \tag{6}$$

where $\mathbf{x}_\theta(s)^\top$ is the row-vector corresponding to state $s$. Further, define the $|\mathcal{S}| \times |\mathcal{S}|$-matrix $P^\pi$ as follows:

$$P^\pi_{s, s'} \triangleq \sum_{a \in \mathcal{A}} \pi(s, a) P(s, a, s'), \quad s, s' \in \mathcal{S}. \tag{7}$$

- Also, recall that $L_{slow}(\theta) = MSTDE(\theta) \triangleq \mathbb{E}\left[\mathbb{E}\left[\delta_t^2 | S_t\right]\right]$.

- A function $\Gamma : U \subseteq \mathbb{R}^{d_1} \to X \subseteq \mathbb{R}^{d_2}$ is Frechet differentiable at $\mathbf{x} \in U$ if there exists a bounded linear operator $\widehat{\Gamma}_{\mathbf{x}} : \mathbb{R}^{d_1} \to \mathbb{R}^{d_2}$ such that the limit

$$\lim_{\epsilon \downarrow 0} \frac{\Gamma(\mathbf{x} + \epsilon \mathbf{y}) - \mathbf{x}}{\epsilon} \tag{8}$$

exists and is equal to $\widehat{\Gamma}_{\mathbf{x}}(\mathbf{y})$. We say $\Gamma$ is Frechet differentiable if Frechet derivative of $\Gamma$ exists at every point in its domain.

## B.2 Assumptions

**Assumption 1:** The pre-determined, deterministic, step-size sequence $\{\xi_t\}_{t \in \mathbb{N}}$ satisfies

$$\xi_t > 0, \forall t \in \mathbb{N}, \quad \sum_{t \in \mathbb{N}} \xi_t = \infty, \quad \sum_{t \in \mathbb{N}} \xi_t^2 < \infty.$$

**Assumption 2:** The Markov chain induced by the given policy $\pi$ is ergodic, *i.e.*, aperiodic and irreducible.

Assumption 2 implies that the underlying Markov chain is asymptotically stationary and henceforth it guarantees the existence of a unique steady-state distribution $\mathbf{d}_\pi$ over the state space $\mathcal{S}$ (Levin and Peres, 2017), *i.e.*, $\lim_{t \to \infty} \mathbb{P}(S_t = s) = \mathbf{d}_\pi(s), \forall s \in \mathcal{S}$.

**Assumption 3:** Given a realization of the transition dynamics of the MDP in the form of a sample trajectory $\mathcal{O}_\pi = \{S_0, A_0, R_1, S_1, A_1, R_2, S_2, \dots\}$, where the initial state $S_0 \in \mathcal{S}$ is chosen arbitrarily, while the action $\mathbb{A} \ni A_t \sim \pi(S_t, \cdot)$, the transitioned state $\mathcal{S} \ni S_{t+1} \sim P(S_t, A_t, \cdot)$ and the reward $\mathbb{R} \ni R_{t+1} = R(S_t, A_t, S_{t+1})$.

## B.3 Convergence Analysis

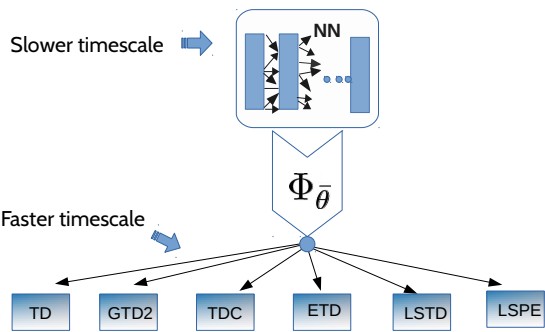

Figure 7: TTN model

To analyze the long-run behaviour of our algorithm, we employ the ODE based analysis (Borkar, 2008; Kushner and Yin, 2003; Ljung, 1977) of the stochastic recursive algorithms. Here, we consider a deterministic ordinary differential equation (ODE) whose asymptotic flow is equivalent to the long-run behaviour of the stochastic recursion. Then we analyze the qualitative behaviour of the solutions of the ODE to determine the asymptotically stable sets. The ODE-based analysis is elegant and conclusive and it further guarantees that the limit points of the stochastic recursion will almost surely belong to the compact connected internally chain transitive invariant set of the equivalent ODE. Since the algorithm follows a multi-timescale stochastic approximation framework, we will also resort to the more generalized multi-timescale differential inclusion based analysis proposed in (Borkar, 1997; Ramaswamy and Bhatnagar, 2016).

Note that there exists only a unilateral coupling between the neural network (where the feature vectors $\bar{\theta}_t$ are calibrated by following a stochastic gradient descent w.r.t. $L_{slow}$) and the various policy evaluation algorithms (see Figure 7). This literally implies that the policy evaluation algorithms depend on the feature vectors $\bar{\theta}_t$ but not vice-versa. Therefore, one can independently analyze the asymptotic behaviour of the feature vectors $\{\bar{\theta}_t\}_{t \in \mathbb{N}}$. Also, as a technical requirement, note that since one cannot guarantee the stability (almost sure boundedness) of the iterates $\{\bar{\theta}_t\}_{t \in \mathbb{N}}$ (which is a necessary condition required for the ODE based analysis. Please refer Chapter 2 of Borkar (2008)), we consider the following projected stochastic recursion:

$$\bar{\theta}_{t+1} = \Gamma^\Theta \left( \bar{\theta}_t + \xi_t \delta_t \left( \nabla_{\bar{\theta}_t} \hat{Y}_{\bar{\theta}}(S_t) - \gamma_{t+1} \nabla_{\bar{\theta}_t} \hat{Y}_{\bar{\theta}}(S_{t+1}) \right) \right), \tag{9}$$

where $\Gamma^\Theta(\cdot)$ is the projection onto a pre-determined compact and convex subset $\Theta \subset \mathbb{R}^{m+d}$, *i.e.*, $\Gamma^\Theta(\mathbf{x}) = \mathbf{x}$, for $\mathbf{x} \in \mathring{\Theta}$, while for $\mathbf{x} \notin \mathring{\Theta}$, it is the nearest point in $\Theta$ w.r.t. the Euclidean distance (or equivalent metric).

Define the filtration $\{\mathcal{F}_t\}_{t \in \mathbb{N}}$, a family of increasing natural $\sigma$-fields, where $\mathcal{F}_t \triangleq \sigma\left(\{\bar{\theta}_i, S_i, R_i; 0 \le i \le t\}\right)$.

The following lemma characterizes the limiting behaviour of the iterates $\{\bar{\theta}_t\}_{t \in \mathbb{N}}$:

**Lemma 1.** *Let Assumptions 1-3 hold. Let $\Theta \subset \mathbb{R}^{m+d}$ be a compact, convex subset with smooth boundary. Let $\Gamma^\Theta$ be Frechet differentiable. Further, let $\widehat{\Gamma}_{\bar{\theta}}^\Theta(-\frac{1}{2}\nabla L_{slow})(\bar{\theta})$ be Lipschitz continuous. Let $\mathcal{K}$ be the set of asymptotically stable equilibria of the following ODE contained inside $\Theta$:*

$$\frac{d}{dt}\bar{\theta}(t) = \widehat{\Gamma}_{\bar{\theta}(t)}^\Theta(-\frac{1}{2}\nabla_{\bar{\theta}}L_{slow})(\bar{\theta}(t)), \quad \bar{\theta}(0) \in \mathring{\Theta} \text{ and } t \in \mathbb{R}_+.$$

*Then the stochastic sequence $\{\bar{\theta}_t\}_{t \in \mathbb{N}}$ generated by the TTN converges almost surely to $\mathcal{K}$.*

*Proof.* We employ here the ODE based analysis as proposed in (Borkar, 2008; Kushner and Clark, 2012). Firstly, we recall here the stochastic recursion which updates $\bar{\theta}_t$:

$$\bar{\theta}_{t+1} = \Gamma^\Theta\left(\bar{\theta}_t + \xi_t \delta_t \left(\nabla_{\bar{\theta}_t}\hat{Y}_{\bar{\theta}}(S_t) - \gamma_{t+1}\nabla_{\bar{\theta}_t}\hat{Y}_{\bar{\theta}}(S_{t+1})\right)\right), \tag{10}$$

where $\Gamma^\Theta$ is the projection onto a pre-determined compact and convex subset $\Theta \subset \mathbb{R}^{m+d}$. Here, $\delta_t \triangleq R_{t+1} + \gamma_{t+1}\hat{Y}_{\bar{\theta}_t}(S_{t+1}) - \hat{Y}_{\bar{\theta}_t}(S_t)$ is the temporal difference. Also, $\nabla_{\bar{\theta}_t}\hat{Y}_{\bar{\theta}} \in \mathbb{R}^{(m+d)\times|\mathcal{S}|}$ is the Jacobian of $\hat{Y}_{\bar{\theta}}$ at $\bar{\theta} = \bar{\theta}_t$ and $\nabla_{\bar{\theta}_t}\hat{Y}_{\bar{\theta}}(s)$ is the column corresponding to state $s$.

Now the above equation can be rewritten as

$$\bar{\theta}_{t+1} = \Gamma^\Theta\left(\bar{\theta}_t + \xi_t\left(h^1(\bar{\theta}_t) + \mathbb{M}_{t+1}^1 + \ell_t^1\right)\right), \tag{11}$$

where $h^1(\bar{\theta}) \triangleq \mathbb{E}\left[\delta_t\left(\nabla_{\bar{\theta}}\hat{Y}_{\bar{\theta}}(S_t) - \gamma_{t+1}\nabla_{\bar{\theta}}\hat{Y}_{\bar{\theta}}(S_{t+1})\right)\right]$, the noise term $\mathbb{M}_{t+1}^1 \triangleq \delta_t\left(\nabla_{\bar{\theta}_t}\hat{Y}_{\bar{\theta}}(S_t) - \gamma_{t+1}\nabla_{\bar{\theta}_t}\hat{Y}_{\bar{\theta}}(S_{t+1})\right) - \mathbb{E}\left[\delta_t\left(\nabla_{\bar{\theta}_t}\hat{Y}_{\bar{\theta}}(S_t) - \gamma_{t+1}\nabla_{\bar{\theta}_t}\hat{Y}_{\bar{\theta}}(S_{t+1})\right)|\mathcal{F}_t\right]$ and the bias $\ell_t^1 \triangleq \mathbb{E}\left[\delta_t\left(\nabla_{\bar{\theta}_t}\hat{Y}_{\bar{\theta}}(S_t) - \gamma_{t+1}\nabla_{\bar{\theta}_t}\hat{Y}_{\bar{\theta}}(S_{t+1})\right)|\mathcal{F}_t\right] - h^1(\bar{\theta}_t)$.

Further,

$$\begin{aligned}
\bar{\theta}_{t+1} &= \bar{\theta}_t + \xi_t\frac{\Gamma^\Theta\left(\bar{\theta}_t + \xi_t\left(h^1(\bar{\theta}_t) + \mathbb{M}_{t+1}^1 + \ell_t^1\right) - \bar{\theta}_t\right)}{\xi_t} \\
&= \bar{\theta}_t + \xi_t\left(\widehat{\Gamma}_{\bar{\theta}_t}^\Theta(h^1(\bar{\theta}_t)) + \widehat{\Gamma}_{\bar{\theta}_t}^\Theta\left(\mathbb{M}_{t+1}^1\right) + \widehat{\Gamma}_{\bar{\theta}_t}^\Theta\left(\ell_t^1\right) + o(\xi_t)\right),
\end{aligned} \tag{12}$$

where $\widehat{\Gamma}^\Theta$ is the Frechet derivative (defined in Eq. (8)). Note that $\Gamma^\Theta$ is single-valued since $\Theta$ is convex and also the above limit exists since the boundary $\partial\Theta$ is assumed smooth. Further, for $\mathbf{x} \in \mathring{\Theta}$, we have

$$\widehat{\Gamma}_{\mathbf{x}}^\Theta(\mathbf{y}) = \lim_{\epsilon \to 0}\frac{\Gamma^\Theta(\mathbf{x} + \epsilon\mathbf{y}) - \mathbf{x}}{\epsilon} = \lim_{\epsilon \to 0}\frac{\mathbf{x} + \epsilon\mathbf{y} - \mathbf{x}}{\epsilon} = \mathbf{y} \text{ (for sufficiently small } \epsilon), \tag{13}$$

*i.e.*, $\widehat{\Gamma}_{\mathbf{x}}^\Theta(\cdot)$ is an identity map for $\mathbf{x} \in \mathring{\Theta}$.

A few observations are in order:

C1: $\widehat{\Gamma}_{\bar{\theta}}^\Theta(h^1(\bar{\theta}))$ is a Lipschitz continuous function in $\bar{\theta}$. This follows from the hypothesis of the Lemma.

C2: $\widehat{\Gamma}_{\bar{\theta}_t}^\Theta\left(\mathbb{M}_{t+1}^1\right)$ is a truncated martingale difference noise. Indeed, it is easy to verify that the noise sequence $\{\mathbb{M}_{t+1}^1\}_{t \in \mathbb{N}}$ is a martingale-difference noise sequence w.r.t to the filtration $\{\mathcal{F}_{t+1}\}_{t \in \mathbb{N}}$, *i.e.*, $\mathbb{M}_{t+1}^1$ is $\mathcal{F}_{t+1}$-measurable and integrable, $\forall t \in \mathbb{N}$ and $\mathbb{E}\left[\mathbb{M}_{t+1}^1|\mathcal{F}_t\right] = 0$ a.s., $\forall t \in \mathbb{N}$. Also, since $\Gamma^\Theta(\cdot)$ is a continuous linear operator, we have $\widehat{\Gamma}^\Theta(\mathbb{M}_{t+1}^1)$ to be $\mathcal{F}_{t+1}$-measurable and integrable, $\forall t \in \mathbb{N}$ likewise.

C3: $\widehat{\Gamma}_{\bar{\theta}_t}^{\Theta}\left(\ell_t^1\right) \to 0$ as $t \to \infty$ a.s. Indeed,

$$\left\|\widehat{\Gamma}_{\bar{\theta}_t}^{\Theta}\left(\ell_t^1\right)\right\| = \left\|\lim_{\epsilon \to 0} \frac{\Gamma^{\Theta}\left(\bar{\theta}_t + \epsilon\ell_t^1\right) - \bar{\theta}_t}{\epsilon}\right\| \le \lim_{\epsilon \to 0} \frac{\left\|\Gamma^{\Theta}\left(\bar{\theta}_t + \epsilon\ell_t^1\right) - \Gamma^{\Theta}\left(\bar{\theta}_t\right)\right\|}{\epsilon}$$

$$\le \lim_{\epsilon \to 0} \frac{\left\|\bar{\theta}_t + \epsilon\ell_t^1 - \bar{\theta}_t\right\|}{\epsilon} = \|\ell_t^1\|.$$

By taking $t \to \infty$, C3 follows directly from the ergodicity (Levin and Peres, 2017) (Assumption 2) and finiteness of the underlying Markov chain.

C4: $o(\xi_t) \to 0$ as $t \to \infty$ (follows from Assumption 1).

C5: Iterates $\{\bar{\theta}_t\}_{t \in \mathbb{N}}$ are stable (forcefully), *i.e.* bounded almost surely, since $\theta_t \in \Theta$, $\forall t \in \mathbb{N}$ (ensured by the projection operator $\Gamma^{\Theta}$) and $\Theta$ is compact (*i.e.*, closed and bounded).

C6: $\exists K_0 \in (0, \infty)$, such that

$$\mathbb{E}\left[\|\widehat{\Gamma}_{\bar{\theta}_t}^{\Theta}\left(\mathbb{M}_{t+1}^1\right)\|^2 | \mathcal{F}_t\right] \le K_0 \left(1 + \|\bar{\theta}_t\|^2\right) \quad a.s. \tag{14}$$

This follows directly from the finiteness of the Markov chain and from the assumption that the boundary $\partial\Theta$ is smooth.

Now, by appealing to Theorem 2, Chapter 2 of (Borkar, 2008)), we conclude that the stochastic recursion (10) asymptotically tracks the following ODE

$$\frac{d}{dt}\bar{\theta}(t) = \widehat{\Gamma}_{\bar{\theta}(t)}^{\Theta}(h^1(\bar{\theta}(t))), \quad \bar{\theta}(0) \in \mathring{\Theta} \text{ and } t \in \mathbb{R}_+$$

$$= \widehat{\Gamma}_{\bar{\theta}(t)}^{\Theta}(-\frac{1}{2}\nabla_{\bar{\theta}}L_{slow})(\bar{\theta}(t)), \quad \bar{\theta}(0) \in \mathring{\Theta} \text{ and } t \in \mathbb{R}_+. \tag{15}$$

In other words, the stochastic recursion (10) converges to the asymptotically stable equilibria of the ODE (15) contained inside $\Theta$.

$\square$

**Remark 1.** *It is indeed non-trivial to determine the constraint set $\Theta$ without prior adequate knowledge about the limit set of the ODE (15). A pragmatic approach to overcome this concern is to initiate the stochastic recursion with an arbitrary convex, compact set $\Theta$ with a smooth boundary and gradually spread to the whole of $\mathbb{R}^{m+d}$ (Chen, 2006).*

**Remark 2.** *It is also important to characterize the hypothesis of the above lemma (i.e., $\widehat{\Gamma}_{\bar{\theta}}^{\Theta}(-\frac{1}{2}\nabla L_{slow})(\bar{\theta})$ is Lipschitz continuous) with respect to the features $\hat{Y}_{\bar{\theta}}$. To achieve that one has to consider the non-projected form of the ODE (15). Apparently, when one considers the spreading approach proposed in the above remark, then it is essentially encouraged to consider the non-projected form since the limiting flow of the ODE arising from the projected stochastic recursion is more likely to lie inside the compact, convex set as $\Theta$ becomes larger. Thereupon, it is easy to observe that the condition $\hat{Y}_{\bar{\theta}}$ is twice continuously-differentiable is sufficient to ensure the Lipschitz continuity of $\widehat{\Gamma}_{\bar{\theta}}^{\Theta}(-\frac{1}{2}\nabla L_{slow})(\bar{\theta})$. Additionally, in that case $\mathcal{K} = \{\bar{\theta}|\nabla_{\bar{\theta}}L_{slow}(\bar{\theta}) = 0\}$ which is the set of local extrema of $J$.*

### B.4 TD($\lambda$) ALGORITHM

One can directly apply the TD($\lambda$) with linear function approximation algorithm to estimate the value function with respect to the features provided by the neural network. The TD($\lambda$) algorithm is provided in Algorithm 2.

Here $\mathbf{e}_t, \mathbf{w}_t \in \mathbb{R}^d$. Further, $\delta_t \triangleq R_{t+1} + \gamma_{t+1}\mathbf{w}_t^{\top}\mathbf{x}_{\theta_t}(S_{t+1}) - \mathbf{w}_t^{\top}\mathbf{x}_{\theta_t}(S_t)$ is the temporal difference.

---

**Algorithm 2** TD($\lambda$) algorithm

---

**Parameters:** $\alpha_t > 0, \lambda \in [0, 1]$;
**Initialization:** $\mathbf{w}_0 = 0, \mathbf{e}_0 = 0$;

For each transition $(S_t, R_{t+1}, S_{t+1})$ in $\mathcal{O}_\pi$, do:

$$\mathbf{e}_{t+1} = \mathbf{x}_{\theta_t}(S_t) + \gamma_{t+1}\lambda \mathbf{e}_t; \tag{16}$$

$$\mathbf{w}_{t+1} = \mathbf{w}_t + \alpha_t \left(R_{t+1} + \gamma_{t+1}\mathbf{w}_t^\top \mathbf{x}_{\theta_t}(S_{t+1}) - \mathbf{w}_t^\top \mathbf{x}_{\theta_t}(S_t)\right)\mathbf{e}_t; \tag{17}$$

---

**Assumption 4-TD($\lambda$):** The pre-determined, deterministic, step-size sequence $\{\alpha_t\}_{t\in\mathbb{N}}$ satisfies:

$$\alpha_t > 0, \forall t \in \mathbb{N}, \quad \sum_{t\in\mathbb{N}}\alpha_t = \infty, \quad \sum_{t\in\mathbb{N}}\alpha_t^2 < \infty, \quad \lim_{t\to\infty}\frac{\xi_t}{\alpha_t} = 0.$$

Note that the step-size schedules $\{\alpha_t\}_{t\in\mathbb{N}}$ and $\{\xi_t\}_{t\in\mathbb{N}}$ satisfy $\frac{\xi_t}{\alpha_t} \to 0$, which implies that $\{\xi_t\}$ converges to 0 relatively faster than $\{\alpha_t\}$. This disparity in terms of the learning rates induces an asynchronous convergence behaviour asymptotically (Borkar, 1997), with feature parameter sequence $\{\bar{\theta}_t\}$ converging slower relative to the TD($\lambda$) sequence $\{\mathbf{w}_t\}$. The rationale being that the increment term of the underlying stochastic gradient descent of the neural network is smaller compared to that of the TD($\lambda$) recursion (17), since the neural network SGD is weighted by the step-size schedule $\{\xi_t\}_{t\in\mathbb{N}}$ which is smaller than $\{\alpha_t\}_{t\in\mathbb{N}}$ for all but finitely many $t$. This unique pseudo heterogeneity induces multiple perspectives, *i.e.*, when viewed from the faster timescale recursion (recursion controlled by $\alpha_t$), the slower timescale recursion (recursion controlled by $\xi_t$) seems quasi-static ('almost a constant'), while viewed from the slower timescale, the faster timescale recursion seems equilibrated. Further, it is analytically admissible (Borkar, 1997) to consider the slow timescale stochastic recursion (*i.e.*, the neural network SGD) to be quasi-stationary (*i.e.*, $\bar{\theta}_t \equiv \bar{\theta}, \forall t \in \mathbb{N}$), while analyzing the asymptotic behaviour of the relatively faster timescale stochastic recursion (17). Thereupon, we obtain the following directly from Theorem 1 of (Tsitsiklis and Van Roy, 1997).

**Lemma 2.** *Assume $\bar{\theta}_t \equiv \bar{\theta}, \forall t \in \mathbb{N}$. Let Assumptions 1-3 and 4-TD($\lambda$) hold. Then for any $\lambda \in [0, 1]$, the stochastic sequence $\{\mathbf{w}_t\}_{t\in\mathbb{N}}$ generated by the TD($\lambda$) algorithm (Algorithm 2) within the TTN setting converges almost surely to the limit $\mathbf{w}^*$, where $\mathbf{w}^*$ satisfies*

$$\Pi_{\bar{\theta}}T^{(\lambda)}(\Phi_{\bar{\theta}}\mathbf{w}^*) = \Phi_{\bar{\theta}}\mathbf{w}^*, \tag{18}$$

*with $T^{(\lambda)}J(s) \triangleq (1 - \lambda)\sum_{i=0}^{\infty}\lambda^i\mathbb{E}\left[\sum_{j=0}^{i}\gamma^{[j]}R_{j+1} + \gamma^{[i+1]}J(S_{i+1})\big|S_0 = s\right]$ and $\gamma^{[j]} = \Pi_{i=0}^{j}\gamma_i$ (with $\gamma_0 = 1$). Also, $\Pi_{\bar{\theta}}$ is defined according to Eq. (3) with $\mathcal{F} = \{\Phi_{\bar{\theta}}\mathbf{w}|\mathbf{w} \in \mathbb{R}^d\}$.*

For other single-timescale prediction methods like ETD and LSPE, similar results follow. Regarding the least squares method LSTD, which offers the significant advantage of non-dependency on step-sizes (albeit computationally expensive) couples smoothly within the TTN setting without any additional consideration.

## B.5 GTD2 Algorithm

However, one cannot directly apply the original GTD2 and TDC algorithms to the TTN setting, since a necessary condition required for the convergence of these algorithms is the non-singularity of the feature specific matrices $\mathbb{E}\left[\mathbf{x}_{\theta_t}(S_t)\mathbf{x}_{\theta_t}(S_t)^\top\right]$ and $\mathbb{E}\left[(\mathbf{x}_{\theta_t}(S_t) - \gamma_{t+1}\mathbf{x}_{\theta_t}(S_{t+1}))\mathbf{x}_{\theta_t}(S_t)^\top\right]$. Please refer Theorem 1 and Theorem 2 of (Sutton et al., 2009). Without the non-singularity assumption, it is indeed hard to guarantee the almost sure boundedness of the GTD2/TDC iterates. In the TTN setting that we consider in this paper, one cannot explicitly assure this condition, since the features are apparently administered by a neural network and it is not directly intuitive on how to control the neural network to generate a collection of features with the desired non-singularity characteristic. Henceforth, one has to consider the projected versions of these algorithms. We consider here the projected GTD2 algorithm provided in Algorithm 3.

---

**Algorithm 3** GTD2 algorithm

---

**Parameters:** $\alpha_t$, $\beta_t$;
**Initialization:** $\mathbf{u}_0 \in U, \mathbf{w}_0 \in W$;

For each transition $(S_t, R_{t+1}, S_{t+1})$ in $\mathcal{O}_\pi$ do:

$$\mathbf{w}_{t+1} = \Gamma^W \left( \mathbf{w}_t + \alpha_t \left( \delta_{t+1}^{\mathbf{u}_t} \mathbf{x}_{\theta_t}(S_t) - \left( \mathbf{w}_t^\top \mathbf{x}_{\theta_t}(S_t) \right) \mathbf{x}_{\theta_t}(S_t) \right) \right); \qquad (19)$$

$$\mathbf{u}_{t+1} = \Gamma^U \left( \mathbf{u}_t + \beta_t \left( \mathbf{x}_{\theta_t}(S_t) - \gamma_{t+1} \mathbf{x}_{\theta_t}(S_{t+1}) \right) \left( \mathbf{w}_t^\top \mathbf{x}_{\theta_t}(S_t) \right) \right); \qquad (20)$$

---

Here $\mathbf{u}_t, \mathbf{w}_t \in \mathbb{R}^d$. Further, $\delta_{t+1}^{\mathbf{u}} \triangleq R_{t+1} + \gamma_{t+1} \mathbf{u}^\top \mathbf{x}_{\theta_t}(S_{t+1}) - \mathbf{u}^\top \mathbf{x}_{\theta_t}(S_t)$ is the temporal difference.

Here, $\Gamma^W(\cdot)$ is the projection operator onto a pre-determined convex, compact subset $W \subset \mathbb{R}^d$ with a smooth boundary $\partial W$. Therefore, $\Gamma^W$ maps vectors in $\mathbb{R}^d$ to the nearest vectors in $W$ w.r.t. the Euclidean distance (or equivalent metric). Convexity and compactness ensure that the projection is unique and belongs to $W$. Similarly, $U$ is a pre-determined convex, compact subset of $\mathbb{R}^d$ with a smooth boundary $\partial U$.

Projection is required since the stability of the iterates $\{\mathbf{w}_t\}_{t \in \mathbb{N}}$ and $\{\mathbf{u}_t\}_{t \in \mathbb{N}}$ are hard to guarantee otherwise.

**Assumption 4-GTD2:** The pre-determined, deterministic, step-size sequences $\{\alpha_t\}_{t \in \mathbb{N}}$ and $\{\beta_t\}_{t \in \mathbb{N}}$ satisfy

$$\alpha_t, \beta_t > 0, \forall t \in \mathbb{N}, \quad \sum_{t \in \mathbb{N}} \alpha_t = \sum_{t \in \mathbb{N}} \beta_t = \infty,$$

$$\sum_{t \in \mathbb{N}} \left( \alpha_t^2 + \beta_t^2 \right) < \infty, \quad \lim_{t \to \infty} \frac{\beta_t}{\alpha_t} = 0, \quad \lim_{t \to \infty} \frac{\xi_t}{\beta_t} = 0.$$

Define the filtration $\{\mathcal{F}_t\}_{t \in \mathbb{N}}$, a family of increasing natural $\sigma$-fields, where $\mathcal{F}_t \triangleq \sigma \left( \{\mathbf{w}_i, \mathbf{u}_i, \bar{\theta}_i, S_i, R_i; 0 \le i \le t\} \right)$.

Similar to the TD($\lambda$) case, here also we follow the quasi-stationary argument. Henceforth, we analyze the asymptotic behaviour of GTD2 algorithm under the assumption that feature vector $\bar{\theta}_t$ is quasi-static, *i.e.* $\bar{\theta}_t \equiv \bar{\theta} = (\theta, \bar{\mathbf{w}})^\top$.

**Lemma 3.** *Assume $\bar{\theta}_t \equiv \bar{\theta} = (\theta, \bar{\mathbf{w}})^\top, \forall t \in \mathbb{N}$. Let Assumptions 1-3 and 4-GTD2 hold. Then*

$$\left\{ (\mathbf{u}, \mathbf{w})^\top \,\big|\, \liminf_{t \to \infty} \left\| (\mathbf{u}, \mathbf{w})^\top - (\mathbf{u}_t, \mathbf{w}_t)^\top \right\| \right\} \subseteq \bigcup_{\mathbf{u} \in \mathcal{A}_*} \left\{ (\mathbf{u}, \mathbf{w})^\top \,\big|\, \mathbf{w} \in \mathcal{A}_{\mathbf{u}} \right\}, \qquad (21)$$

*where $\mathcal{A}_*$ is the set of asymptotically stable equilibria of the following ODE:*

$$\frac{d}{dt} \mathbf{u}(t) = \widehat{\Gamma}_{\mathbf{u}(t)}^U \left( \Phi_{\bar{\theta}}^\top D_{\mathbf{d}_\pi} (\mathbb{I} - \gamma_{t+1} P^\pi) \Phi_{\bar{\theta}} \mathbf{u}(t) \right), \quad \mathbf{u}(0) \in \mathring{U}, \ t \in \mathbb{R}_+ \qquad (22)$$

*and $\mathcal{A}_{\mathbf{u}}$ is the asymptotically stable equilibria of the following ODE:*

$$\frac{d}{dt} \mathbf{w}(t) = \widehat{\Gamma}_{\mathbf{w}(t)}^W \left( \left( \Phi_{\bar{\theta}}^\top D_{\mathbf{d}_\pi} \delta^{\mathbf{u}} - \Phi_{\bar{\theta}}^\top D_{\mathbf{d}_\pi} \Phi_{\bar{\theta}} \right) \mathbf{w}(t) \right), \quad \mathbf{w}(0) \in \mathring{W} \text{ and } t \in \mathbb{R}_+,$$

*with $\delta^{\mathbf{u}}$ defined in Eq. (29).*

*Proof.* The two equations in the modified GTD2 algorithm constitute a multi-timescale stochastic approximation recursion, where there exists a bilateral coupling between the stochastic recursions (19) and (20). Since the step-size sequences $\{\alpha_t\}_{t \in \mathbb{N}}$ and $\{\beta_t\}_{t \in \mathbb{N}}$ satisfy $\frac{\beta_t}{\alpha_t} \to 0$, we have $\beta_t \to 0$ faster than $\alpha_t \to 0$. This disparity in terms of the learning rates induces a pseudo heterogeneous rate of convergence (or timescales) between the individual stochastic recursions which results in a

pseudo asynchronous convergence behaviour when considered over a finite time window. Also note that the coherent long-run behaviour of the multi-timescale stochastic recursion will asymptotically follow this short-term behaviour with the window size extending to infinity(Borkar, 1997; 2008). This pseudo behaviour induces multiple viewpoints, *i.e.*, when observed from the faster timescale recursion (recursion controlled by $\alpha_t$), the slower timescale recursion (recursion controlled by $\beta_t$) appears quasi-static ('almost a constant'), while observed from the slower timescale, the faster timescale recursion seems equilibrated. Further, it is analytically admissible (Borkar, 1997) to consider the slow timescale stochastic recursion (20) to be quasi-stationary (*i.e.*, $\mathbf{u}_t \equiv \mathbf{u}, \forall t \in \mathbb{N}$), while analyzing the limiting behaviour of the relatively faster timescale stochastic recursion 19.

**Analysis of the faster time-scale recursion:** The faster time-scale stochastic recursion of the GTD2 algorithm is the following:

$$\mathbf{w}_{t+1} = \Gamma^W \left( \mathbf{w}_t + \alpha_t \left( \delta_{t+1}^{\mathbf{u}_t} \mathbf{x}_{\theta_t}(S_t) - \left( \mathbf{w}_t^\top \mathbf{x}_{\theta_t}(S_t) \right) \mathbf{x}_{\theta_t}(S_t) \right) \right). \tag{23}$$

Under the previously mentioned quasi-stationary premise that $\mathbf{u}_t \equiv \mathbf{u}$ and $\bar{\theta}_t \equiv \bar{\theta} = (\theta, \bar{w})^\top, \forall t \in \mathbb{N}$, thereupon, we analyze the long-term behaviour of the following recursion:

$$\mathbf{w}_{t+1} = \Gamma^W \left( \mathbf{w}_t + \alpha_t \left( \delta_{t+1}^{\mathbf{u}} \mathbf{x}_t - \left( \mathbf{w}_t^\top \mathbf{x}_t \right) \mathbf{x}_t \right) \right), \tag{24}$$

where $\mathbf{x}_t = \mathbf{x}_\theta(S_t)$ and $\delta_{t+1}^{\mathbf{u}} \triangleq R_{t+1} + \gamma_{t+1} \mathbf{u}^\top \mathbf{x}_{t+1} - \mathbf{u}^\top \mathbf{x}_t$.

The above equation can be rearranged as the following:

$$\mathbf{w}_{t+1} = \Gamma^W \left( \mathbf{w}_t + \alpha_t \left( h^2(\mathbf{w}_t) + \mathbb{M}_{t+1}^2 + \ell_t^2 \right) \right),$$

where the noise $\mathbb{M}_{t+1}^2 \triangleq \delta_{t+1}^{\mathbf{u}} \mathbf{x}_t - \left( \mathbf{w}_t^\top \mathbf{x}_t \right) \mathbf{x}_t - \mathbb{E} \left[ \delta_{t+1}^{\mathbf{u}} \mathbf{x}_t - \left( \mathbf{w}_t^\top \mathbf{x}_t \right) \mathbf{x}_t | \mathcal{F}_t \right]$, $h^2(\mathbf{w}) \triangleq \mathbb{E} \left[ \delta_{t+1}^{\mathbf{u}} \mathbf{x}_t - \left( \mathbf{w}_t^\top \mathbf{x}_t \right) \mathbf{x}_t \right]$ and the bias $\ell_t^2 = \mathbb{E} \left[ \delta_{t+1}^{\mathbf{u}} \mathbf{x}_t - \left( \mathbf{w}_t^\top \mathbf{x}_t \right) \mathbf{x}_t | \mathcal{F}_t \right] - \mathbb{E} \left[ \delta_{t+1}^{\mathbf{u}} \mathbf{x}_t - \left( \mathbf{w}_t^\top \mathbf{x}_t \right) \mathbf{x}_t \right]$.

Similar to Equation (12), we can rewrite the above recursion as follows:

$$\mathbf{w}_t = \mathbf{w}_t + \alpha_t \left( \widehat{\Gamma}_{\mathbf{w}_t}^W(h^2(\mathbf{w}_t)) + \widehat{\Gamma}_{\mathbf{w}_t}^W \left( \mathbb{M}_{t+1}^2 \right) + \widehat{\Gamma}_{\mathbf{w}_t}^W \left( \ell_t^2 \right) + o(\alpha_t) \right), \tag{25}$$

where $\widehat{\Gamma}_{\mathbf{w}_t}^W(\cdot)$ is the Frechet derivative (defined in Equation (8)) of the projection operator $\Gamma^W$.

A few observations are in order:

D1: The iterates $\{\mathbf{w}_t\}_{t \in \mathbb{N}}$ are stable, *i.e.*, $\sup_{t \in \mathbb{N}} \|\mathbf{w}_t\| < \infty$ *a.s.* This immediately follows since $W$ is bounded.

D2: $\{\widehat{\Gamma}_{\mathbf{w}_t}^W \left( \mathbb{M}_{t+1}^2 \right)\}_{t \in \mathbb{N}}$ is a martingale-difference noise sequence with respect to the filtration $\{\mathcal{F}_{t+1}\}_{t \in \mathbb{N}}$. This follows directly since $\{\mathbb{M}_{t+1}^2\}_{t \in \mathbb{N}}$ is a martingale-difference noise sequence with respect to the same filtration.

D3: $\{\widehat{\Gamma}_{\mathbf{w}_t}^W \left( \mathbb{M}_{t+1}^2 \right)\}_{t \in \mathbb{N}}$ are square-integrable and $\exists K_2 \in (0, \infty)$ such that

$$\mathbb{E} \left[ \|\widehat{\Gamma}_{\mathbf{w}_t}^W \left( \mathbb{M}_{t+1}^2 \right)\|^2 | \mathcal{F}_t \right] \leq K_2 \left( 1 + \|\mathbf{w}_t\|^2 \right) \quad a.s., \quad t \in \mathbb{N}. \tag{26}$$

This follows directly from the finiteness of the underlying Markov chain and from the assumption that the boundary $\partial W$ is smooth.

D4: $\widehat{\Gamma}_{\mathbf{w}}^W(h^2(\mathbf{w}))$ is Lipschitz continuous with respect to $\mathbf{w}$. Proof similar to C1.

D5: $\widehat{\Gamma}_{\mathbf{w}_t}^W \left( \ell_t^2 \right) \to 0$ as $t \to \infty$ *a.s.* Proof similar to C3.

Now by appealing to Theorem 2, Chapter 2 of (Borkar, 2008) along with the above observations, we conclude that the stochastic recursion 23 asymptotically tracks the following ODE almost surely:

$$\frac{d}{dt} \mathbf{w}(t) = \widehat{\Gamma}_{\mathbf{w}(t)}^W(h^2(\mathbf{w}(t)), \quad \mathbf{w}(0) \in \mathring{W} \text{ and } t \in \mathbb{R}_+.$$

$$= \widehat{\Gamma}_{\mathbf{w}(t)}^W \left( \mathbb{E} \left[ \delta_{t+1}^{\mathbf{u}} \mathbf{x}_t \right] - \mathbb{E} \left[ \mathbf{x}_t \mathbf{x}_t^\top \right] \mathbf{w}(t) \right), \quad \mathbf{w}(0) \in \mathring{W} \text{ and } t \in \mathbb{R}_+. \tag{27}$$

Therefore, $\mathbf{w}_t$ converges asymptotically to the stable equilibria of the above ODE contained inside $W$ almost surely.

**Qualitative analysis of the solutions of ODE (27)**: A trivial qualitative analysis of the long-run behaviour of the flow induced by the above ODE attests that the stable limit set is indeed the solutions of the following linear system inside $W$ (This follows since $\widehat{\Gamma}_{\mathbf{w}}^{W}(\mathbf{y}) = \mathbf{y}$ for $\mathbf{w} \in \mathring{W}$ and also because $\widehat{\Gamma}_{\mathbf{w}}^{W}(\cdot)$ does not contribute any additional limit points on the boundary other than the roots of $h^2$ since $\partial W$ is smooth).

$$\mathbb{E}\left[\mathbf{x}_t \mathbf{x}_t^\top\right] \mathbb{E}\left[\delta_{t+1}^{\mathbf{u}} \mathbf{x}_t\right] - \mathbb{E}\left[\mathbf{x}_t \mathbf{x}_t^\top\right] \mathbb{E}\left[\mathbf{x}_t \mathbf{x}_t^\top\right] \mathbf{w} = 0.$$
$$\Rightarrow \mathbb{E}\left[\mathbf{x}_t \mathbf{x}_t^\top\right] \mathbf{w} = \mathbb{E}\left[\delta_{t+1}^{\mathbf{u}} \mathbf{x}_t\right]. \tag{28}$$

Note that $\mathbb{E}\left[\mathbf{x}_t \mathbf{x}_t^\top\right] = \Phi_{\bar{\theta}}^\top D_{\mathbf{d}_\pi} \Phi_{\bar{\theta}}$.

**Claim 1:** The above linear system of equations is consistent, *i.e.*, $\mathbb{E}\left[\delta_{t+1}^{\mathbf{u}} \mathbf{x}_t\right] \in \mathcal{R}(\Phi_{\bar{\theta}}^\top D_{\mathbf{d}_\pi} \Phi_{\bar{\theta}})$, *i.e.*, the range-space of $\Phi_{\bar{\theta}}^\top D_{\mathbf{d}_\pi} \Phi_{\bar{\theta}}$: To see that, note that the above system can indeed be viewed as the least squares solution to the $\Phi_{\bar{\theta}} \mathbf{w} = \delta^{\mathbf{u}}$ with respect to the weighted-norm $\|\cdot\|_{D_{\mathbf{d}_\pi}}$, where

$$\delta^{\mathbf{u}}(s) = \bar{R}^\pi(s) + \gamma_{t+1} \sum_{s' \in \mathcal{S}} P_{s,s'}^\pi \mathbf{u}^\top \mathbf{x}_\theta(s') - \sum_{s' \in \mathcal{S}} P_{s,s'}^\pi \mathbf{u}^\top \mathbf{x}_\theta(s'), \tag{29}$$

where $\bar{R}$ is the expected reward. (Note that $\mathbb{E}\left[\delta_{t+1}^{\mathbf{u}} \mathbf{x}_t\right] = \Phi_{\bar{\theta}}^\top D_{\mathbf{d}_\pi} \delta^{\mathbf{u}}$).

The least-squares solution $\mathbf{w}_0 \in \mathbb{R}^d$ (which certainly exists but may not be unique) satisfies

$$< \Phi_{\bar{\theta}} \mathbf{w}, \delta^{\mathbf{u}} - \Phi_{\bar{\theta}} \mathbf{w}_0 >_{D_{\mathbf{d}_\pi}} = 0, \quad \forall \mathbf{w} \in \mathbb{R}^d$$
$$\Rightarrow \ < \mathbf{w}, D_{\mathbf{d}_\pi}^{-1} \Phi_{\bar{\theta}}^\top D_{\mathbf{d}_\pi} (\delta^{\mathbf{u}} - \Phi_{\bar{\theta}} \mathbf{w}_0) >_{D_{\mathbf{d}_\pi}} = 0, \quad \forall \mathbf{w} \in \mathbb{R}^d.$$

Now choose $\mathbf{w} = D_{\mathbf{d}_\pi}^{-1} \Phi_{\bar{\theta}}^\top D_{\mathbf{d}_\pi} (\delta^{\mathbf{u}} - \Phi_{\bar{\theta}} \mathbf{w}_0)$. Then

$$\Phi_{\bar{\theta}}^\top D_{\mathbf{d}_\pi} (\delta^{\mathbf{u}} - \Phi_{\bar{\theta}} \mathbf{w}_0) = 0 \ \Rightarrow \ \Phi_{\bar{\theta}}^\top D_{\mathbf{d}_\pi} \Phi_{\bar{\theta}} \mathbf{w}_0 = \Phi_{\bar{\theta}}^\top D_{\mathbf{d}_\pi} \delta^{\mathbf{u}}. \quad \text{[ End of proof of Claim 1 ]}$$

Since $\Phi_{\bar{\theta}}^\top D_{\mathbf{d}_\pi} \Phi_{\bar{\theta}}$ may be singular (*i.e.*, $\Phi_{\bar{\theta}}^\top D_{\mathbf{d}_\pi} \Phi_{\bar{\theta}}$ is not invertible), the above least squares solution may not be unique and hence the collection of asymptotically stable equilibria of the flow induced by the ODE (27) may not be singleton for every $\mathbf{u}$. Let's denote the asymptotically stable equilibria of the flow induced by the said ODE to be the set $\mathcal{A}_{\mathbf{u}}$, where $\emptyset \neq \mathcal{A}_{\mathbf{u}} \subseteq W$.

**Analysis of the slower time-scale recursion:** The slower time-scale stochastic recursion of the GTD2 algorithm is the following:

$$\mathbf{u}_{t+1} = \Gamma^U \left(\mathbf{u}_t + \beta_t \left(\mathbf{x}_t - \gamma_{t+1} \mathbf{x}_{t+1}\right) \left(\mathbf{w}_t^\top \mathbf{x}_t\right)\right), \quad \mathbf{u}_t \in \mathbb{R}^d, \quad \mathbf{u}_0 \in U. \tag{30}$$

Note that since $\frac{\xi_t}{\beta_t} \to 0$, the stochastic recursion (20) is managed on a faster timescale relative to the the neural network stochastic recursion (10) and henceforth, we continue to maintain here the quasi-stationary condition $\bar{\theta}_t \equiv \bar{\theta} = (\theta, \bar{w})^\top$.

Now the above equation can be rearranged as follows:

$$\mathbf{u}_{t+1} = \Gamma^U \left(\mathbf{u}_t + \beta_t \left(\mathbb{E}\left[\Delta_{t+1}^{\mathbf{w}_t}\right] + \mathbb{M}_{t+1}^3 + \ell_t^3\right)\right), \tag{31}$$

where $\Delta_{t+1}^{\mathbf{w}_t} \triangleq \left(\mathbf{x}_t - \gamma_{t+1} \mathbf{x}_{t+1}\right) \left(\mathbf{w}_t^\top \mathbf{x}_t\right) = \left(\left(\mathbf{x}_t - \gamma_{t+1} \mathbf{x}_{t+1}\right) \mathbf{x}_t^\top\right) \mathbf{w}_t$, the noise term $\mathbb{M}_{t+1}^3 \triangleq \Delta_{t+1}^{\mathbf{w}_t} - \mathbb{E}\left[\Delta_{t+1}^{\mathbf{w}_t} | \mathcal{F}_t\right]$ and the bias $\ell_t^3 \triangleq \mathbb{E}\left[\Delta_{t+1}^{\mathbf{w}_t} | \mathcal{F}_t\right] - \mathbb{E}\left[\Delta_{t+1}^{\mathbf{w}_t}\right]$.

Similar to Equation (12), we can rewrite the above recursion as follows:

$$\mathbf{u}_{t+1} = \mathbf{u}_t + \beta_t \left(\widehat{\Gamma}_{\mathbf{u}_t}^{U} \left(\mathbb{E}\left[\Delta_{t+1}^{\mathbf{w}_t}\right]\right) + \widehat{\Gamma}_{\mathbf{u}_t}^{U} \left(\mathbb{M}_{t+1}^3\right) + \widehat{\Gamma}_{\mathbf{u}_t}^{U} \left(\ell_t^3\right) + o(\beta_t)\right), \tag{32}$$

where $\widehat{\Gamma}_{\mathbf{u}_t}^{U}(\cdot)$ is the Frechet derivative (defined in Equation (8)) of the projection operator $\Gamma^U$.

Now the above equation can be interpreted in terms of stochastic recursive inclusion as follows:

$$\mathbf{u}_{t+1} = \mathbf{u}_t + \beta_t \left(\widehat{\Gamma}_{\mathbf{u}_t}^{U} \left(\mathbb{E}\left[\Delta_{t+1}^{\mathbf{w}_t}\right]\right) + \widehat{\Gamma}_{\mathbf{u}_t}^{U} \left(\mathbb{M}_{t+1}^3\right) + \widehat{\Gamma}_{\mathbf{u}_t}^{U} \left(\ell_t^3\right) + o(\beta_t)\right), \tag{33}$$

with $\widehat{\Gamma}^U_{\mathbf{u}_t}\left(\mathbb{E}\left[\Delta^{\mathbf{w}_t}_{t+1}\right]\right) \in h^3(\mathbf{u}_t)$, where the set-valued map $h^3 : \mathbb{R}^d \to \{\text{subsets of } \mathbb{R}^d\}$ is defined as

$$h^3(\mathbf{u}) \triangleq \left\{\widehat{\Gamma}^U_{\mathbf{u}}\left(\mathbb{E}\left[\Delta^{\mathbf{w}}_{t+1}\right]\right), \text{ where } \mathbf{w} \in \mathcal{A}_{\mathbf{u}}\right\}. \tag{34}$$

Indeed $h^3(\mathbf{u}) = \left\{\widehat{\Gamma}^U_{\mathbf{u}}\left(B\mathbf{w}\right), \text{ where } B = \mathbb{E}\left[\left((\mathbf{x}_t - \gamma_{t+1}\mathbf{x}_{t+1})\,\mathbf{x}^\top_t\right)\right] \text{ and } \mathbf{w} \in \mathcal{A}_{\mathbf{u}}\right\}$. It is easy to verify that $B = \Phi^\top_{\bar{\theta}} D_{\mathbf{d}_\pi}(\mathbb{I} - \gamma_{t+1}P^\pi)\Phi_{\bar{\theta}}$.

Here, one cannot directly apply the multi-timescale stochastic approximation results from (Borkar, 1997) since the said paper assumes that the limit point from the slower timescale recursion is unique (Please see Chapter 6 of (Borkar, 2008)). But in our setting, the slower timescale recursion (23) has several limit points (note that the stable limit set $\mathcal{A}_{\mathbf{u}}$ is not singleton). This is where our analysis differs from that of the seminal paper on GTD2 algorithm, where it is assumed that both the matrices $\mathbb{E}\left[\mathbf{x}_t\mathbf{x}^\top_t\right]$ and $\mathbb{E}\left[(\mathbf{x}_t - \gamma_{t+1}\mathbf{x}_{t+1})\mathbf{x}^\top_t\right]$ are certainly non-singular. However, in our TTN setting, one cannot guarantee this condition, since the features are apparently provided by a neural network and it is hard to fabricate the neural network to generate a collection of features with the desired non-singularity properties. In order to analyze the limiting behaviour of the GTD2 algorithm under the relaxed singularity setting, henceforth one has to view the stochastic recursion (30) as a stochastic recursion inclusion (Benaïm et al., 2005) and apply the recent results from (Ramaswamy and Bhatnagar, 2016) which analyzes the asymptotic behaviour of general multi-timescale stochastic recursive inclusions.

A few observations are in order:

E1: For each $\mathbf{u} \in U$, $h^3(\mathbf{u})$ is a singleton: This follows from the definition of $h^3$ and Claim 1 above, where we established that each $\mathbf{w} \in \mathcal{A}_{\mathbf{u}}$ is the least squares solution to the linear system of equations $\Phi_{\bar{\theta}}\mathbf{w} = \delta^{\mathbf{u}}$. Therefore, it further implies that that $h^3$ is a Marchaud map as well.

E2: $\sup_{t\in\mathbb{N}}\left(\|\mathbf{w}_t\| + \|\mathbf{u}_t\|\right) < \infty$ a.s. This follows since $W$ and $U$ are bounded sets.

E3: $\{\widehat{\Gamma}^U_{\mathbf{u}_t}\left(\mathbb{M}^3_{t+1}\right)\}_{t\in\mathbb{N}}$ is a martingale-difference noise sequence with respect to the filtration $\{\mathcal{F}_{t+1}\}_{t\in\mathbb{N}}$. This follows directly since $\{\mathbb{M}^3_{t+1}\}_{t\in\mathbb{N}}$ is a martingale-difference noise sequence with respect to the same filtration.

E4: $\{\widehat{\Gamma}^U_{\mathbf{u}_t}\left(\mathbb{M}^3_{t+1}\right)\}_{t\in\mathbb{N}}$ are square-integrable and $\exists K_3 \in (0, \infty)$, such that

$$\mathbb{E}\left[\left\|\widehat{\Gamma}^U_{\mathbf{u}_t}\left(\mathbb{M}^3_{t+1}\right)\right\|^2 \Big| \mathcal{F}_t\right] \le K_3\left(1 + \|\mathbf{u}_t\|^2 + \|\mathbf{w}_t\|^2\right) \quad a.s., \quad t \in \mathbb{N}. \tag{35}$$

This follows directly from the finiteness of the underlying Markov chain and from the assumption that the boundary $\partial U$ is smooth.

E5: $\widehat{\Gamma}^U_{\mathbf{u}_t}\left(\ell^3_t\right) \to 0$ as $t \to \infty$ a.s. Proof similar to C3. This implies that the bias is asymptotically irrelevant.

E6: For each $\mathbf{u} \in U$, the set $\mathcal{A}_{\mathbf{u}}$ is a globally attracting set of the ODE (27) and is also Lyapunov stable. Further, there exists $K_4 \in (0, \infty)$ such that $\sup_{\mathbf{w}\in\mathcal{A}_{\mathbf{u}}}\|\mathbf{w}\| \le K_4(1 + \|\mathbf{u}\|)$. This follows since $\mathcal{A}_{\mathbf{u}} \subseteq W$ and $W$ is bounded.

E7: The set-valued map $q : U \to \{\text{subsets of } \mathbb{R}^d\}$ given by $q(\mathbf{u}) = \mathcal{A}_{\mathbf{u}}$ is upper-semicontinuous: Consider the convergent sequences $\{\mathbf{u}_n\}_{n\in\mathbb{N}} \to \mathbf{u}$ and $\{\mathbf{w}_n\}_{n\in\mathbb{N}} \to \mathbf{w}$ with $\mathbf{u}_n \in U$ and $\mathbf{w}_n \in q(\mathbf{u}_n) = \mathcal{A}_{\mathbf{u}}$. Note that $\mathbf{w} \in W$, $\mathbf{u} \in U$ since $W$ and $U$ are compact. Also $\Phi^\top_{\bar{\theta}} D_{\mathbf{d}_\pi}\Phi_{\bar{\theta}}\mathbf{w}_n = \Phi^\top_{\bar{\theta}} D_{\mathbf{d}_\pi}\delta^{\mathbf{u}_n}$ (from Claim 1). Now taking limits on both sides we get

$$\lim_{n\to\infty}\Phi^\top_{\bar{\theta}} D_{\mathbf{d}_\pi}\Phi_{\bar{\theta}}\mathbf{w}_n = \lim_{n\to\infty}\Phi^\top_{\bar{\theta}} D_{\mathbf{d}_\pi}\delta^{\mathbf{u}_n} \quad \Rightarrow \quad \Phi^\top_{\bar{\theta}} D_{\mathbf{d}_\pi}\Phi_{\bar{\theta}}\mathbf{w} = \Phi^\top_{\bar{\theta}} D_{\mathbf{d}_\pi}\delta^{\mathbf{u}}.$$

This implies that $\mathbf{w} \in \mathcal{A}_{\mathbf{u}} = q(\mathbf{u})$. The claim thus follows.

Thus we have established all the necessary conditions demanded by Theorem 3.10 of (Ramaswamy and Bhatnagar, 2016) to characterize the limiting behaviour of the stochastic recursive inclusion (33).

Now by appealing to the said theorem, we obtain the following result on the asymptotic behaviour of the GTD2 algorithm:

$$\left\{ (\mathbf{u}, \mathbf{w})^{\top} \,\middle|\, \liminf_{t \to \infty} \left\| (\mathbf{u}, \mathbf{w})\top - (\mathbf{u}_t, \mathbf{w}_t)^{\top} \right\| \right\} \subseteq \bigcup_{\mathbf{u} \in \mathcal{A}_*} \left\{ (\mathbf{u}, \mathbf{w})^{\top} \,\middle|\, \mathbf{w} \in \mathcal{A}_{\mathbf{u}} \right\}, \tag{36}$$

where $\mathcal{A}_*$ is the set of asymptotically stable equilibria of the following ODE:

$$\frac{d}{dt} \mathbf{u}(t) = h^3(\mathbf{u}(t)), \quad \mathbf{u}(0) \in \mathring{U}, \ t \in \mathbb{R}_+. \tag{37}$$

$\square$

One can obtain similar results for projected TDC.

We now state our main result:

**Theorem 2.** *Let $\Theta \subset \mathbb{R}^{m+d}$ be a compact, convex subset with smooth boundary. Let $\Gamma^{\Theta}$ be Frechet differentiable. Further, let $\widehat{\Gamma}^{\Theta}_{\bar{\theta}}(-\frac{1}{2}\nabla L_{slow})(\bar{\theta})$ be Lipschitz continuous. Also, let Assumptions 1-3 hold. Let $\mathcal{K}$ be the set of asymptotically stable equilibria of the following ODE contained inside $\Theta$:*

$$\frac{d}{dt} \bar{\theta}(t) = \widehat{\Gamma}^{\Theta}_{\bar{\theta}(t)}(-\frac{1}{2}\nabla_{\bar{\theta}} L_{slow})(\bar{\theta}(t)), \quad \bar{\theta}(0) \in \mathring{\Theta} \text{ and } t \in \mathbb{R}_+.$$

*Then the stochastic sequence $\{\bar{\theta}_t\}_{t \in \mathbb{N}}$ generated by the TTN converges almost surely to $\mathcal{K}$ (sample path dependent). Further,*

*TD($\lambda$) Convergence: Under the additional Assumption 4-TD($\lambda$), we obtain the following result: For any $\lambda \in [0, 1]$, the stochastic sequence $\{\mathbf{w}_t\}_{t \in \mathbb{N}}$ generated by the TD($\lambda$) algorithm (Algorithm 2) within the TTN setting converges almost surely to the limit $\mathbf{w}^*$, where $\mathbf{w}^*$ satisfies*

$$\Pi_{\bar{\theta}^*} T^{(\lambda)}(\Phi_{\bar{\theta}^*} \mathbf{w}^*) = \Phi_{\bar{\theta}^*} \mathbf{w}^*, \tag{38}$$

*with $T^{(\lambda)}$ defined in Lemma 2 and $\bar{\theta}^* \in \mathcal{K}$ (sample path dependent).*

*GTD2 Convergence: Let $W, U \subset \mathbb{R}^d$ be compact, convex subsets with smooth boundaries. Let Assumption 4-GTD2 hold. Let $\Gamma^W$ and $\Gamma^U$ be Frechet differentiable. Then the stochastic sequences $\{\mathbf{w}_t\}_{t \in \mathbb{N}}$ and $\{\mathbf{u}_t\}_{t \in \mathbb{N}}$ generated by the GTD2 algorithm (Algorithm 3) within the TTN setting satisfy*

$$\left\{ (\mathbf{u}, \mathbf{w})^{\top} \,\middle|\, \liminf_{t \to \infty} \left\| (\mathbf{u}, \mathbf{w})^{\top} - (\mathbf{u}_t, \mathbf{w}_t)^{\top} \right\| \right\} \subseteq \bigcup_{\mathbf{u} \in \mathcal{A}_*} \left\{ (\mathbf{u}, \mathbf{w})^{\top} \,\middle|\, \mathbf{w} \in \mathcal{A}_{\mathbf{u}} \right\},$$

*where $\mathcal{A}_*$ is the set of asymptotically stable equilibria of the following ODE:*

$$\frac{d}{dt} \mathbf{u}(t) = \widehat{\Gamma}^U_{\mathbf{u}(t)} \left( \Phi_{\bar{\theta}^*}^{\top} D_{\mathbf{d}_{\pi}}(\mathbb{I} - \gamma_{t+1} P^{\pi}) \Phi_{\bar{\theta}^*} \mathbf{u}(t) \right), \quad \mathbf{u}(0) \in \mathring{U}, \ t \in \mathbb{R}_+$$

*and $\mathcal{A}_{\mathbf{u}}$ is the asymptotically stable equilibria of the following ODE:*

$$\frac{d}{dt} \mathbf{w}(t) = \widehat{\Gamma}^W_{\mathbf{w}(t)} \left( \left( \Phi_{\bar{\theta}^*}^{\top} D_{\mathbf{d}_{\pi}} \delta^{\mathbf{u}} - \Phi_{\bar{\theta}^*}^{\top} D_{\mathbf{d}_{\pi}} \Phi_{\bar{\theta}^*} \right) \mathbf{w}(t) \right), \quad \mathbf{w}(0) \in \mathring{W} \text{ and } t \in \mathbb{R}_+,$$

*with $\bar{\theta}^* \in \mathcal{K}$ (sample path dependent) and $\delta^{\mathbf{u}}$ defined in Eq. (29).*

# C ADDITIONAL EXPERIMENTS

## C.1 NONIMAGE CATCHER

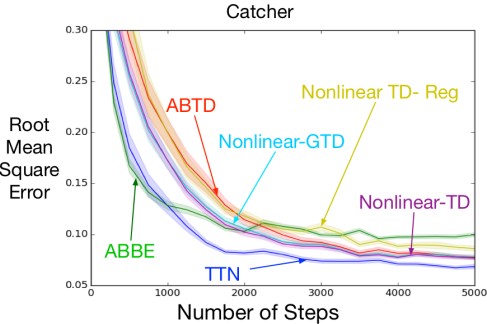

Figure 8: TTN comparison to other nonlinear value function approximation algorithms. We use LSTD for the fast part of the TTN.

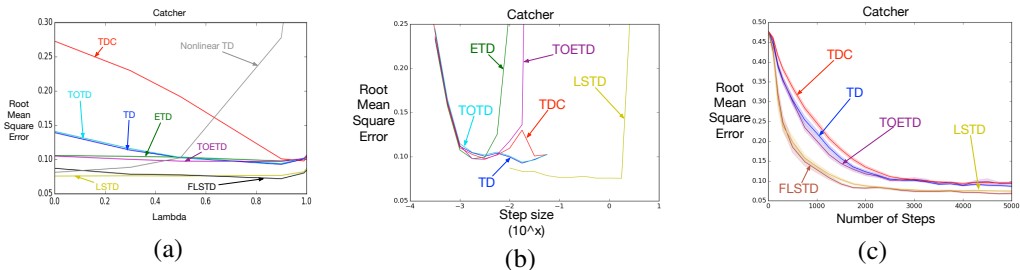

(a)  (b)  (c)

Figure 9: a) Sensitivity plots for $\lambda \in \{0, 0.3, 0.5, 0.9, 0.99, 1\}$ b) Step size sensitivities. c) Comparison of least-squares methods

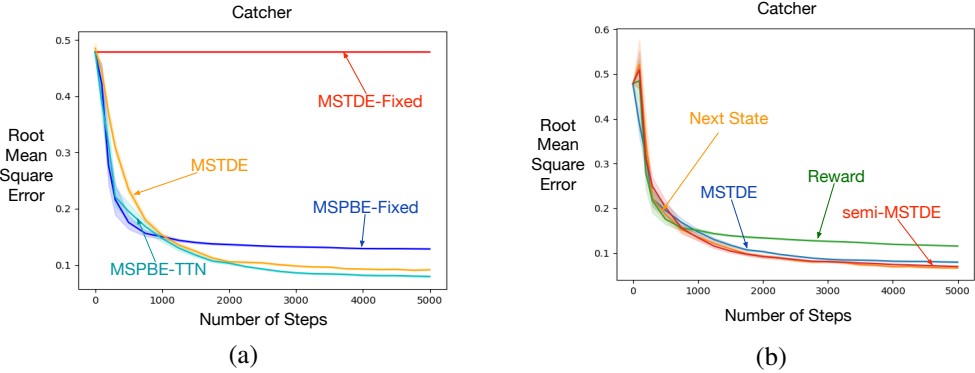

(a)  (b)

Figure 10: a) Comparison of MSPBE and MSTDE. b) Comparison of surrogate loss functions.

## C.2 PUDDLEWORLD

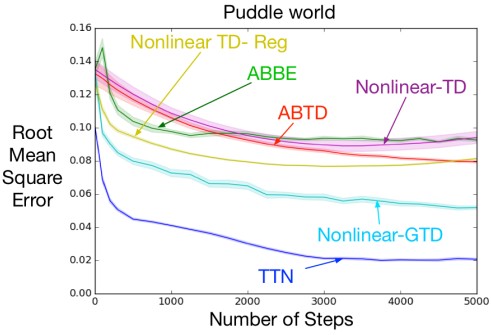

Figure 11: TTN comparison to other nonlinear value function approximation algorithms. We use LSTD for the fast part of the TTN.

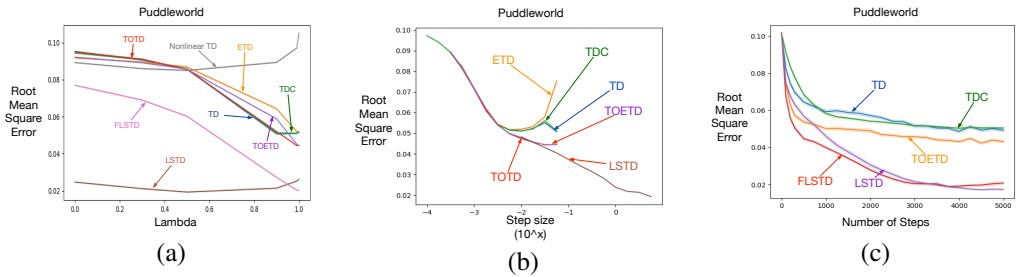

Figure 12: a) Sensitivity plots for $\lambda \in \{0, 0.3, 0.5, 0.9, 0.99, 1\}$ b) Step size sensitivities. c) Comparison of least-squares methods

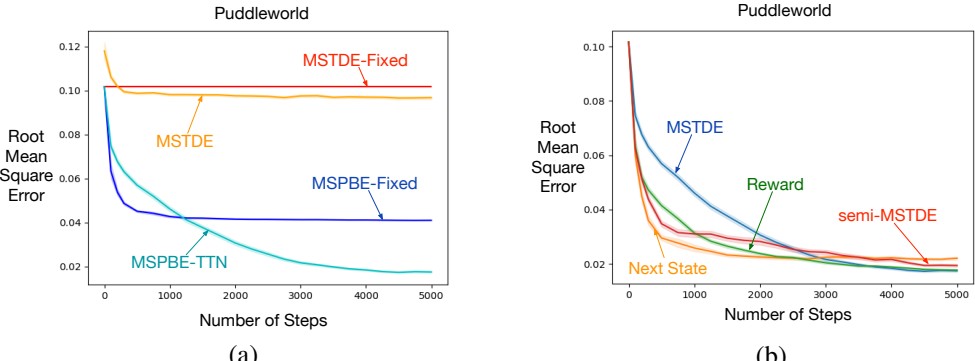

Figure 13: a) Comparison of MSPBE and MSTDE. b) Comparison of surrogate loss functions.

## C.3 IMAGE CATCHER

We also ran policy evaluation experiments on image-based catcher with 2 stacked 64x64 frames as input. The policy evaluated was the same as was used in the non-image setting. Similar to the non-imaged based catcher experiments, we have similar plots.

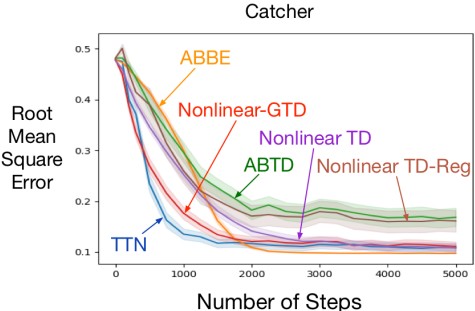

Figure 14: TTN comparison to other nonlinear value function approximation algorithms. We use LSTD for the fast part of the TTN.

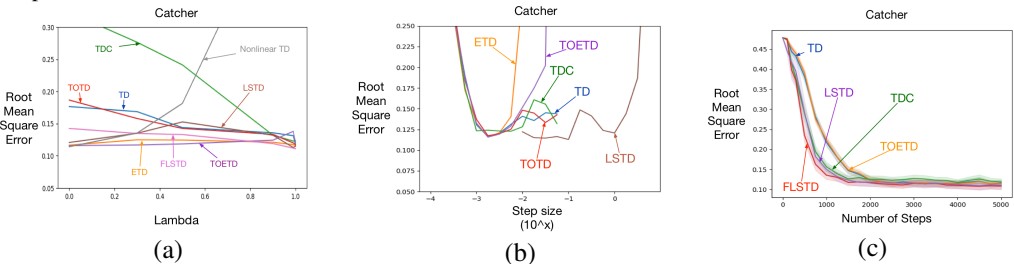

Figure 15: a) Sensitivity plots for $\lambda \in \{0, 0.3, 0.5, 0.9, 0.99, 1\}$ b) Step size sensitivities. c) Comparison of least-squares methods

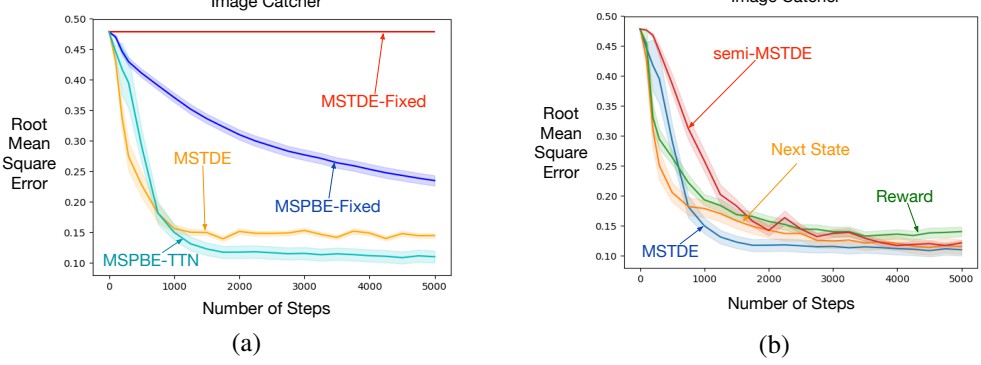

Figure 16: a) Comparison of MPSBE and MSTDE. b) Comparison of surrogate loss functions.

## C.4 CARTPOLE

In the classic Cartpole environment, the agent has to balance a pole on a cart. The state is given by vector of 4 numbers (cart position, cart velocity, pole angle, pole velocity). The two available actions are applying a force towards the left or the right. Rewards are +1 at every timestep and an episode terminates once the pole dips below a certain angle or the cart moves too far from the center. We use the OpenAI gym implementation (Brockman et al., 2016).

The policy to be evaluated consists of applying force in the direction the pole is moving with probability 0.9 (stabilizing the pole) or applying force in the direction of the cart's velocity with probability 0.1. We inject some stochasticity so that the resulting policy does not perform overly well, which would lead to an uninteresting value function.

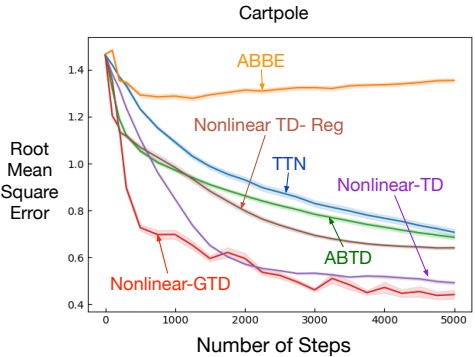

Figure 17: TTN comparison to other nonlinear value function approximation algorithms. We use LSTD for the fast part of the TTN.

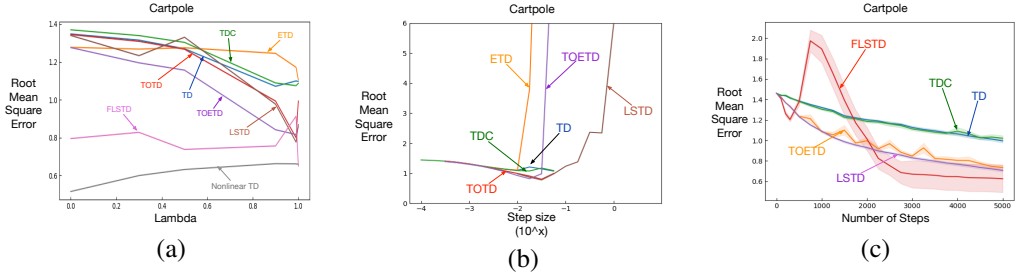

Figure 18: a) Sensitivity plots for $\lambda \in \{0, 0.3, 0.5, 0.9, 0.99, 1\}$ b) Step size sensitivities. c) Comparison of least-squares methods

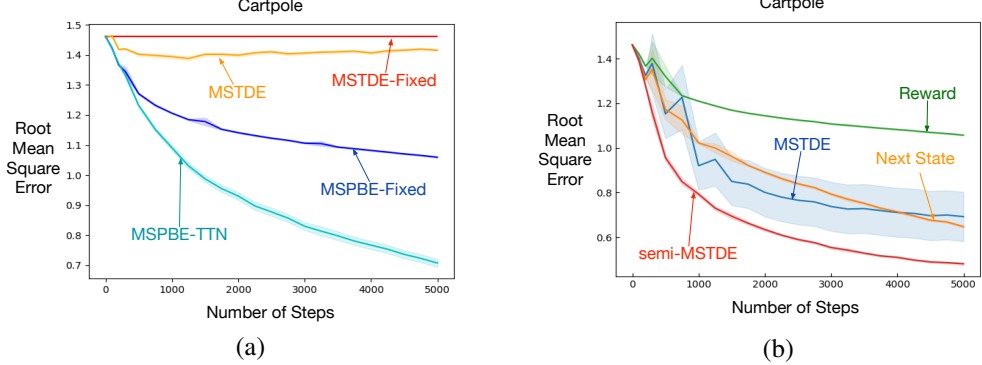

Figure 19: a) Comparison of MSPBE and MSTDE. b) Comparison of surrogate loss functions.

### C.5 ACROBOT

In the classic Acrobot domain, the agent consisting of two links has to swing up past a certain height. The agent observes a 4-dimensional state consisting of the angles and the angular velocities of each link. The avaiable actions are three possible levels of torque to be applied to the joint.

The evaluated policy is obtained by training an agent with true-online Sarsa on a tile coding representation and then fixing its learned epsilon-greedy policy.

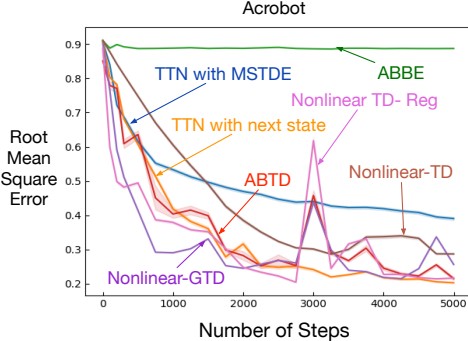

Figure 20: TTN comparison to other nonlinear value function approximation algorithms. We use LSTD for the fast part of the TTN.

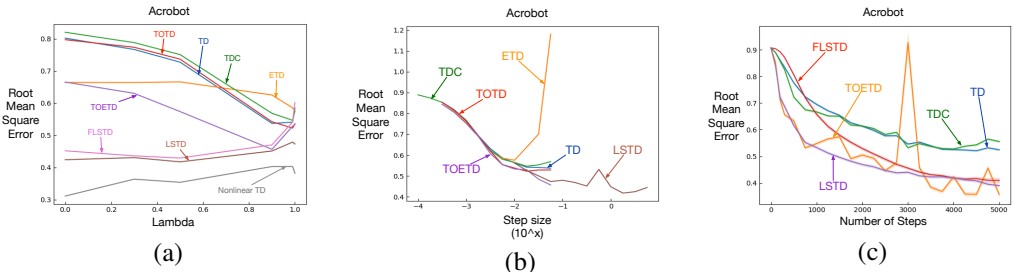

(a)                                  (b)                                  (c)

Figure 21: a) Sensitivity plots for $\lambda \in \{0, 0.3, 0.5, 0.9, 0.99, 1\}$ b) Step size sensitivities. c) Comparison of least-squares methods

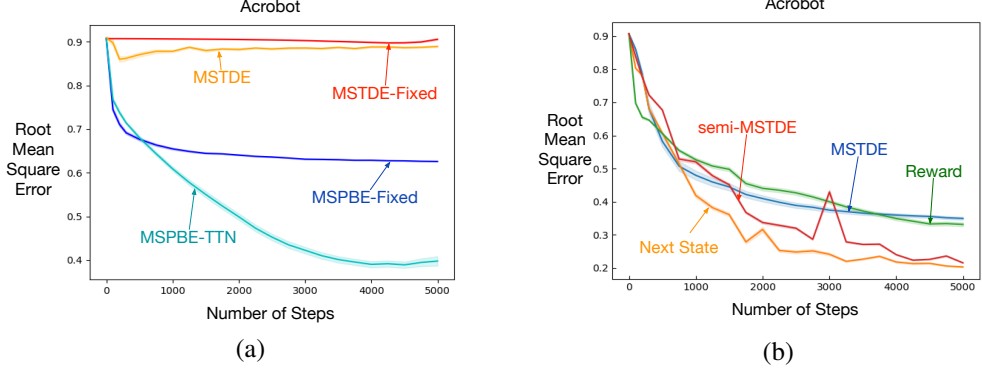

(a)                                  (b)

Figure 22: a) Comparison of Comparison of MSPBE and MSTDE. b) Comparison of surrogate loss functions.

## C.6 PUCK WORLD

In Puck World (Tasfi, 2016), the agent has to move in a two-dimensional box towards a good puck while staying away from a bad puck. The 8-dimensional state consists of (player x location, player y location, player x velocity, player y velocity, good puck x location, good puck y location, bad puck x location, bad puck y location). Each action increases the agent's velocity in one of the four cardinal directions apart from a "None" action which does nothing. The reward is the negative distance to the good puck plus a penalty of $-10 + x$ if the agent is within a certain radius of the bad puck, where $x \in [-2, 0]$ depends on the distance to the bad puck (the reward is slightly modified from the original game to make the value function more interesting).

The policy moves the agent towards the good puck, while having a soft cap on the agent's velocity. In more detail, to choose one action, it is defined by the following procedure: First, we choose some *eligible* actions. The None action is always eligible. The actions which move the agent towards the good puck are also eligible. For example, if the good puck is Northeast of the agent, the North and East actions are eligible. If the agent's velocity in a certain direction is above 30, then the action for that direction is no longer eligible. Finally, the agent picks uniformly at random from all eligible actions.

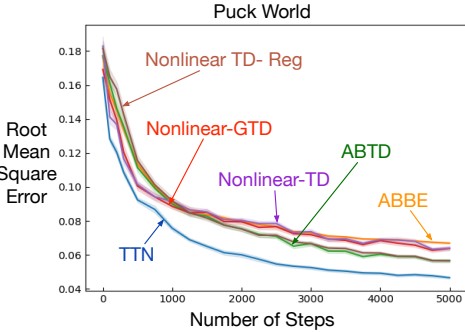

Figure 23: TTN comparison to other nonlinear value function approximation algorithms. We use LSTD for the fast part of the TTN.

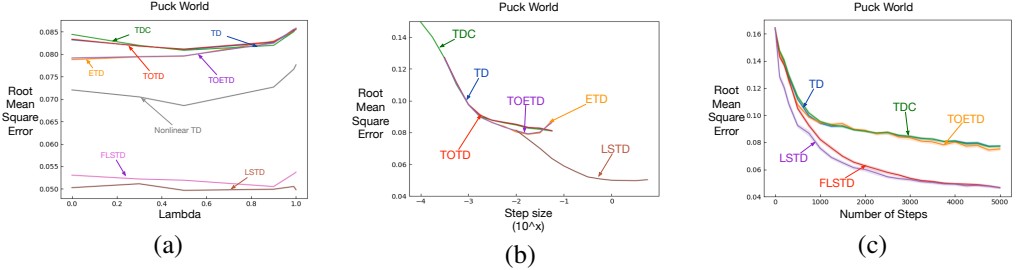

Figure 24: a) Sensitivity plots for $\lambda \in \{0, 0.3, 0.5, 0.9, 0.99, 1\}$ b) Step size sensitivities. c) Comparison of least-squares methods

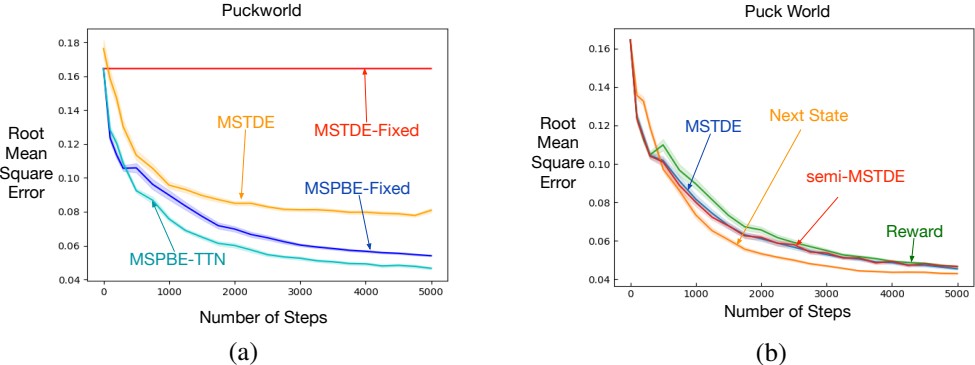

Figure 25: a) Comparison of MSPBE and MSTDE. b) Comparison of surrogate loss functions.

### C.7 OFF-POLICY CATCHER

We run a preliminary experiment to check if TTN can have an advantage in the off-policy setting. The target policy is the same as the one used for other Catcher experiments (described in Appendix D).

The behaviour policy is slightly different. If the apple is within 20 units (the target policy is 25 units), then the agent takes the action in the direction of the apple with probability 0.7 and one of the other two actions with probability 0.15 each. If the apple is not within range, then the agent takes the None action 10% of the time and one of the other two with equal probability. This combination of behaviour and target policies results in importance sampling ratios in the range of 0 to 8.7, moderately large values.

We try TTN with three off-policy algorithms (TD, TDC and LSTD) and compare to off-policy Nonlinear TD. For TTN, the features are learnt optimizing the MSTDE on the behaviour policy while the values are learned off-policy. The main difference between TTN and Nonlinear TD is the fact that Nonlinear TD does off-policy updates to the entire network while TTN only changes the linear part with off-policy updates.

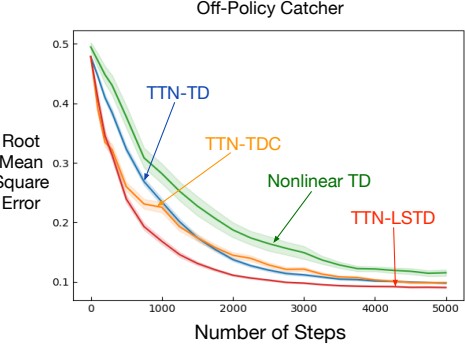

Figure 26: Comparison of TTN and nonlinear TD

From figure C.7, we see that TTN can outperform nonlinear TD in terms of average error and also has reduced variance (smaller error bars). This seems to suggest that the TTN architecture can grant additional stability in the off-policy setting.

## D EXPERIMENTAL DETAILS

**LSTD**  This version of LSTD was used in the policy evaluation experiments.

---
**Algorithm 4** Incremental LSTD algorithm

---

**Inputs:** $\eta_{inv}, \lambda, w_1$ (initial weight vector)
**Initialization:** $A_1 = diag(\eta_{inv}) \in \mathbb{R}^{d \times d}, b_1 = 0 \in \mathbb{R}^d, z_1 = 0 \in \mathbb{R}^d$, $d$ is the number of features

For each transition $(S_t, R_{t+1}, S_{t+1})$ do:
$$\Delta_t = (x(S_t) - \gamma x(S_{t+1}))^T$$
$$A_{t+1} = \frac{t+1}{t} \left( A_t - \frac{A_t z_t \Delta_t A_t}{t + \Delta_t A_t z_t} \right)$$
$$b_{t+1} = \frac{1}{t+1}(z_t * r_t - b_t)$$
$$w_{t+1} = A_{t+1} b_{t+1}$$
$$z_{t+1} = \lambda \gamma z_t + x(S_t)$$

---

**Description of policies**  The policy evaluated in Puddle World randomly took the north and east actions with equal probability, with episodes initialized in the southwest corner. The policy evaluated

in Catcher increased the velocity in the direction of the apple if the agent was within 25 units or else chose the "None" action with a 20% chance of a random choice.

**Competitor details**   Nonlinear TD uses the semi-gradient TD update with nonlinear function approximation. This is known not to be theoretically-sound and there exist counterexamples where the weights and predictions diverge (Tsitsiklis and Van Roy, 1997). Nevertheless, since this is the simplest algorithm, we use this as a baseline. Nonlinear GTD, conversely, has proven convergence results as it is an extension of the gradient TD methods to nonlinear function approximation.

Castro et al. proposed three variants of adaptive bases algorithms, each optimizing a different objective. We include two of them: ABTD, which is based on the plain TD update and ABBE, which is derived from the MSTDE. We omit ABPBE, which optimizes the MSPBE, since the algorithm is computationally inefficient, requiring $O(d^2 m)$ memory, where $d$ is the number of features in the last layer and $m$ is the number of parameters for the bases (ie. weights in the neural network). Also, the derivation is similar in spirit to that of nonlinear GTD, so we would expect both to perform similarly.

Levine et al.'s algorithm was proposed for the control setting, combining DQN with periodic LSTD re-solving for the last layer's weights. We adapt their idea for policy evaluation by adding regularization to the last layer's weights to bias them towards the LSTD solution and using semi-gradient TD on this objective. Then, we can then train in a fully online manner, which makes the algorithm comparable to the other competitors. This algorithm is labeled as Nonlinear TD-Reg.

**Hyperparameters**   Here we present the refined hyperparameter ranges that were swept over for the experimental runs.

First, we outline the hyperparameters of all the algorithms and, afterwards, we give the values tested for each hyperparameter. There is one range for Puddle World and another for Catcher (both versions) since different ranges worked well for each domain. For the two-timescale networks, for each environment, the learning rate for the slow part was set to a fixed value which was sufficiently low for the fast part to do well but was not tuned extensively. This was done for all the experiments except for those on surrogate losses, where we swept over a range of values (since different losses could have different optimization properties).

Two-timescale Networks:
For all algorithms, the learning rate for the slow part $\alpha_{slow}$.
TD, ETD, TOTD, TOETD : learning rate $\alpha$, trace parameter $\lambda$
TDC : primary weights learning rate $\alpha$, secondary weights learning rate $\beta$, trace parameter $\lambda$
LSTD: initializer for the $A^{-1}$ matrix $\eta_{inv}$, trace parameter $\lambda$
FLSTD: initializer for the $A$ matrix $\eta$, learning rate $\alpha$, forgetting rate $\psi$, trace parameter $\lambda$

Castro MSTDE, Castro TD: final layer learning rate $\alpha$, other layers learning rate $\beta$

Nonlinear GTD : primary weights learning rate $\alpha$, secondary weights learning rate $\beta$

Nonlinear TD : learning rate $\alpha$, trace parameter $\lambda$

Nonlinear TD - Reg : learning rate $\alpha$, learning rate towards LSTD solution (regularization) $\beta$, initializer for the A matrix $\eta$

For all algorithms that used eligibility traces, the range swept over was $\lambda \in \{0, 0.3, 0.5, 0.9, 0.99, 1\}$ The other hyperparameters are shown individually below.

Two-timescale Networks:

$$\alpha_{slow} = 10^{-2.5}$$

TD, ETD, TOTD, TOETD:

$$\alpha \in 10^c,\ c \in \{-3.5, -3.25, ..., -1.25\}$$

TDC:

$$\alpha \in 10^c,\ c \in \{-4, -2.75, ..., -1.25\}$$

$$\beta \in 10^c,\ c \in \{-3, -2\}$$

LSTD:
$$\eta_{inv} \in 10^c,\ c \in \{-2, -1.75, ..., 0.75\}$$

FLSTD:
$$\eta \in 10^c,\ c \in \{-3, -2, -1, 0\}$$
$$\alpha \in 10^c,\ c \in \{-4, -3\}$$
$$\psi \in 10^c,\ c \in \{-5, -4, -3\}$$

Castro MSTDE, Castro TD:
$$\alpha \in 10^c,\ c \in \{-6, -5.75, ..., -4\}$$
$$\beta \in 10^c,\ c \in \{-6, -5.75, ..., -4\}$$

Nonlinear GTD:
$$\alpha \in 10^c,\ c \in \{-3, -2.75, ..., -0.25\}$$
$$\beta \in 10^c,\ c \in \{-5, -4.5, ..., -0.5\}$$

Nonlinear TD:
$$\alpha \in 10^c,\ c \in \{-4, -3.75, ..., -1.25\}$$

Nonlinear TD - Reg:
$$\alpha \in 10^c,\ c \in \{-6, -5.5, ..., -3\}$$
$$\beta \in 10^c,\ c \in \{-3.5, -3, ..., -0.5\}$$
$$\eta \in 10^c,\ c \in \{-2, -1, 0, 1\}$$

**Control experiments** For both control experiments, we modified the Catcher environment slightly from its default settings. The agent is given only 1 life and an episode terminates after it fails to catch a single apple. The reward for catching an apple is +1 and is -1 at the end of an episode. The discount factor is set to 0.99.

For image-based Catcher, we stack two consecutive frames which we treat as the state. This is done so that the agent can perceive movements in the paddle, thus making the state Markov.

Both DQN and TTN use an $\epsilon$-greedy policy. For DQN, $\epsilon$ is annealed from 1.0 to 0.01 (0.1) for nonimage (image) Catcher over a certain number of steps. For TTN, $\epsilon$ is fixed to a constant value, 0.01 (0.1) for nonimage (image) Catcher.

For both algorithms, we use a replay buffer of size 50000 (200000) for nonimage (image) Catcher. The buffer is initialized with 5000 (50000) transitions from a random policy. The minibatch size for DQN and feature learning was 32. TTN uses the AMSGrad optimizer with $\beta_1 = 0$ and $\beta_2 = 0.99$ and DQN uses the ADAM optimizer with default settings, $\beta_1 = 0.9, \beta_2 = 0.999$.

For image catcher, due to the long training times, the hyperparameters were manually tuned. For nonimage catcher, we concentrated our hyperparameter tuning efforts on the most important ones and use the following strategy. We first used a preliminary search to find a promising range of values, followed by a grid search. For TTN with FQI, we focused tuning on the step sizes for the features and the regularization factor.For DQN, we focused on adjusting the step size and the number of steps over which $\epsilon$, the probability of picking a random action, was annealed. The other hyperparameters were left to reasonable values.

The final hyperparameters for nonimage catcher:

TTN — $\alpha_{slow} = 10^{-3}$, $\lambda_{reg} = 10^{-2}$ (regularization towards previous weights)

DQN — $\alpha = 10^{-3.75}$, decaying $\epsilon$ over 20 thousand steps, update the target network every 1000 steps. For LS-DQN, do a FQI upate every 50,000 steps with a regularization weight of 1.

The final hyperparameters for image catcher:

TTN — $\alpha_{slow} = 10^{-5}$, $\lambda_{reg} = 10^{-3}$ (regularization towards previous weights), solve for new weights using FQI every 10,000 steps.

DQN — $\alpha = 10^{-3.75}$, decaying $\epsilon$ over 1 million steps, update the target network every 10,000 steps. For LS-DQN, do a FQI update every 500,000 steps with a regularization weight of 1.

*Network architectures*
For TTN, this describes the neural network which learns the features. The layer marked as "features" is used to predict state-action values for the fast, linear part.

Image catcher - TTN

To predict the next state, we add an action embedding as done in (Oh et al., 2015).

| | |
|---|---|
| Input | 2x64x64 grayscale image (stacked frames) |
| Conv1 | 32 filters, 7x7 kernel, stride 4, ReLU |
| Conv2 | 64 filters, 5x5 kernel, stride 2, ReLU |
| Conv3 | 64 filters, 3x3 kernel, stride 1, ReLU |
| Dense1 (features) | 256 units, ReLU |
| From Dense1, Dense2 (reward prediction) | 3 units (one per action), Linear |
| From Dense1, add action embedding | 256 units, Linear |
| Reshape | Into 8x8 image |
| Deconv1 | 16 filters, 5x5 kernel, stride 2, ReLU |
| Deconv2 | 16 filters, 5x5 kernel, stride 2, ReLU |
| Deconv3 | 16 filters, 5x5 kernel, stride 2, ReLU |
| Conv4 (state prediction) | 1 filter, 3x3 kernel, stride 1, Linear |

Image catcher - DQN

| | |
|---|---|
| Input | 2x64x64 grayscale image (stacked frames) |
| Conv1 | 32 filters, 7x7 kernel, stride 4, ReLU |
| Conv2 | 64 filters, 5x5 kernel, stride 2, ReLU |
| Conv3 | 64 filters, 3x3 kernel, stride 1, ReLU |
| Dense1 | 256 units, ReLU |
| Dense2 (Q-values) | 3 units, Linear |

Nonimage catcher - TTN

| | |
|---|---|
| Input | 4-dimensional state |
| Dense1 | 128 units, ReLU |
| Dense2 | 128 units, ReLU |
| Dense3 (features) | 128 units, ReLU |
| Dense4 (state and reward prediction) | 15 units, Linear |

Nonimage catcher - DQN

| | |
|---|---|
| Input | 4-dimensional state |
| Dense1 | 128 units, ReLU |
| Dense2 | 128 units, ReLU |
| Dense3 | 128 units, ReLU |
| Dense4 (Q-values) | 3, Linear |

