# OpenReview forum: "Two-Timescale Networks for Nonlinear Value Function Approximation"
_ICLR.cc/2019/Conference_

### Official Review · AnonReviewer2 · 2018-11-04
**A well written paper with thorough experimental evaluation, but lacks novelty.**

**Rating:** 6
**Confidence:** 4

**Review:**

Summary:
This paper presents a Two-Timescale Network (TTN) that enables linear methods to be used to learn values. On the slow timescale non-linear features are learned using a surrogate loss. On the fast timescale, a value function is estimated as a linear function of those features. It appears to be a single network, where one head drives the representation and the second head learns the values.  They investigate multiple surrogate losses and end up using the MSTDE for its simplicity, even though it provides worse value estimates than MSPBE as detailed in their experiments.  They provide convergence results - regular two-timescale stochastic approximation results from Borkar, for the two-timescale procedure and provide empirical evidence for the benefits of this method compared to other non-linear value function approximation methods.

Clarity and Quality:
The paper is well written in general, the mathematics seems to be sound and the experimental results appear to be thorough.

Originality:
Using two different heads, one to drive the representation and the second to learn the values appears to be an architectural detail. The surrogate loss to learn the features coupled with a linear policy evaluation algorithm appear to be novel, but does not warrant, in my opinion, the novelty necessary for publication at ICLR.

The theoretical results appear to be a straightforward application of Borkar’s two-timescale stochastic approximation algorithm to this architecture to get convergence. This therefore, does not appear to be a novel contribution.

You state after equaltion (3) that non-linear function classes do not have a closed form solution. However, it seems that the paper Convergent Temporal-Difference Learning with Arbitrary Smooth Function Approximation does indeed have a closed form solution for non-linear function approximators when minimizing the MSPBE (albeit making a linearity assumption, which is something your work seems to make as well).

The work done in the control setting appears to be very similar to the experiments performed in the paper: Shallow Updates for Deep Reinforcement Learning.

Significance:
Overall, I think that the paper is well written and the experimental evaluation is thorough. However, the novelty is lacking as it appears to be training using a multi-headed approach (which exists) and the convergence results appear to be a straightforward application of Borkars two-timescale proof. The novelty therefore appears to be using a surrogate loss function for training the features which does not possess the sufficient novelty in my opinion for ICLR.

I would suggest the authors' detail why their two-timescale approach is different from that of Borkars. Or additionally add some performance guarantee to the convergence results to extend the theory. This would make for a much stronger paper.

---

> ### Author Response · Authors · 2018-11-08
> **Thank you for the review**
>
> Thank you for your constructive feedback and comments. We look forward to further discussion.
>
> Concerning the novelty of the algorithm, we would like to emphasize that the main focus of the two-timescale network (TTN) architecture is to separate the value and feature learning processes. This is in contrast to the more popular approach of jointly learning values and features in an end-to-end manner. Splitting these learning processes is key to providing convergence guarantees and enabling the use of advances for linear policy evaluation which do not have direct extension to nonlinear function approximation, such as least-squares methods and eligibility traces. Empirically, we show that these additions can provide benefits for policy evaluation. We also validate that TTNs can still achieve competitive performance in the control setting even when the features are learned separately, unlike the architecture that is used by Levine et al. (“Shallow Updates for Deep Reinforcement Learning”) where the features are learned jointly.
>
> We would like to clarify our contributions in regards to the theoretical analysis.
> The analysis provided in the paper has two parts. In the first part (Lemma 1), we address the convergence of the feature representation algorithm with Markovian noise (contrary to the IID setting), where we employed extensions of the classical results from Borkar’s textbook.
> In the second part (Lemma 3), we address the value function prediction procedure, where we found that Borkar’s classical results do not apply due to the singularity of the feature covariance matrix involved. These singular matrices can occur since we do not assume that the feature-learning process produces linearly independent features, an unrealistic assumption for neural networks. This renders the analysis more complex and non-trivial. We address this issue by considering the algorithm as a multi-timescale stochastic approximation *inclusion* (contrary to a stochastic approximation recursion) and employ recent results on multi-timescale stochastic approximation inclusions from “A. Ramaswamy and S. Bhatnagar. Stochastic recursive inclusion in two timescales with an application to the lagrangian dual problem. Stochastics, 2016.” Hence the proof techniques employed are totally different and novel in their application to reinforcement learning.
> Regarding the performance guarantees suggested by the reviewer, we agree that these would be highly desired but, currently, it is an open problem to develop sample complexity bounds for general multi-timescale stochastic approximation algorithms. The only result we are aware of applies solely to linear systems (Konda and Tsitsiklis, “Convergence rate of two-time-scale stochastic approximation”, Annals of Applied Probability, 2004). Additionally, our algorithm is a multi-timescale stochastic approximation *inclusion* which would pose an even larger challenge.
>
> Concerning “...the paper Convergent Temporal-Difference Learning with Arbitrary Smooth Function Approximation does indeed have a closed form solution for non-linear function approximators…”
> Indeed, Maei et al. are able to find a closed-form solution for the projection operator onto a tangent plane approximation. Without this approximation, the projection operator would not have a readily-available form. Our main point is that, despite this simplifying assumption, the projection operator still depends on the current parameter values, which complicates the derivation and results in a more complex algorithm (Nonlinear GTD).

---

> > ### Comment · AnonReviewer2 · 2018-11-23
> > **Backtracking on my novelty comment**
> >
> > Thank you for clarifying the novelty. I have adjusted my score accordingly.
> > However, I urge the authors to clarify the theoretical novelty in their paper and include a sketch proof in the main paper to provide intuition as to why inclusions are necessary.

---

### Official Review · AnonReviewer3 · 2018-11-13
**A paper with a lot of potential but not well structured. I suggest to rewrite it for a journal track.**

**Rating:** 6
**Confidence:** 4

**Review:**

The paper proposes a two-timescale framework for learning the value function and a state representation altogether with nonlinear approximators. The authors provide proof of convergence and a good empirical evaluation.

The topic is very interesting and relevant to ICLR. However, I think that the paper is not ready for a publication.
First, although the paper is well written, the writing can be improved. For instance, I found already the abstract a bit confusing. There, the authors state that they "provide a two-timescale network (TTN) architecture that enables LINEAR methods to be used to learn values [...] The approach facilitates use of algorithms developed for the LINEAR setting [...] We prove convergence for TTNs, with particular care given to ensure convergence of the fast LINEAR component."
Yet, the title says NONLINEAR and in the remainder of the paper they use neural networks.

The major problem of the paper is, however, its organization. The novelty of the paper (the proof of convergence) is relegated to the appendix, and too much is spent in the introduction, when actually the idea of having the V-function depending on a slowly changing network is also not novel in RL. For instance, the authors say that V depends on \theta and w, and that \theta changes at slower pace compared to w. This recalls the use of target networks in the TD error for many actor-critic algorithms. (It is not the same thing, but there is a strong connection).
Furthermore, in the introduction, the authors say that eligibility traces have been used only with linear function approximators, but GAE by Schulman et al. uses the same principle (their advantage is actually the TD(\lambda) error) to learn an advantage function estimator, and it became SOTA for learning the value function.

I am also a bit skeptical about the use of MSBE in the experiment. First, in Eq 4 and 5 the authors state that using the MSTDE is easier than MSBE, then in the experiments they evaluate both. However, the MSBE error involves the square of an expectation, which should be biased. How do you compute it?
(Furthermore, you should spend a couple of sentences to explain the problem of this square and the double-sampling problem of Bellman residual algorithms. For someone unfamiliar with the problem, this issue could be unclear.)

I appreciate the extensive evaluation, but its organization can also be improved, considering that some important information are, again, in the appendix.
Furthermore, results on control experiment are not significative and should be removed (at the current stage, at least). In the non-image version there is a lot of variance in your runs (one blue curve is really bad), while for the image version all runs are very unstable, going always up and down.

In conclusion, there is a lot of interesting material in this paper. Even though the novelty is not great, the proofs, analysis and evaluation make it a solid paper. However, because there is so much do discuss, I would suggest to reorganize the paper and submit directly to a journal track (the paper is already 29 pages including the appendix).

---

> ### Author Response · Authors · 2018-11-24
> **Thank you for the review**
>
> Thank you for your constructive feedback.
>
> We have edited the abstract to clarify that we are indeed considering nonlinear value function approximation by using a combination of nonlinear (learned) features and a linear value function of those features.
>
> Although target networks and the TTN ideas have similar goals---improving the stability of learning nonlinear value functions---the approaches are fairly different. Target networks attempt to stabilize the learning process by fixing the TD targets for some number of steps. On the other hand, TTNs provide stability by ‘fixing’ (slowing down) the change in the features. These two approaches are orthogonal and can definitely be combined. Also, note that the use target networks alone does not provide any convergence guarantees with nonlinear function approximation unlike TTNs.
>
> To clarify, we did not use the MSBE in any of the experiments. Indeed, the double-sampling problem makes its use infeasible in practice and we mention this under equation (6) on page 4. As such, the MSTDE was the surrogate loss of choice for most of the experiments.
>
> We want to clarify that GAE (Schulman et al.) use an analog to the lambda-return with advantage functions, but they do not use eligibility traces. Instead, they use the forward view to compute the desired quantities by accumulating a batch of data. To the best of our knowledge, there are no theoretically-sound extensions of eligibility traces (ie. the backward view) for nonlinear function approximation.
>
> For the control experiments, it is true that the runs of TTN---and DQN---have relatively high variance. We agree that these results definitely leave room for improvement but our goal is to give some preliminary results to show that TTNs can be a promising direction for the control setting in addition to policy evaluation. Further, such variability in control is not unique to this paper, and is a larger research question in RL.

---

> > ### Comment · AnonReviewer3 · 2018-11-25
> > **Response to rebuttal**
> >
> > Thank you for answering my questions. I have adjusted my score accordingly.
> > I suggest you to add few sentences to clarify the novelty, as you explained to me in your response (especially about the eligibility traces and the target/convergence).
> > Also, I would suggest to move the catcher experiments to the appendix unless you can get better results. They are interesting, but not that meaningful at the current stage. You can use the space to move some text from the appendix to the main part, such as Algorithm 1.

---

### Official Review · AnonReviewer4 · 2018-11-14
**Interesting algorithm, although similar methods and claims have been proposed recently**

**Rating:** 7
**Confidence:** 4

**Review:**

This paper proposes Two-Timescale Networks (TTNs), a reinforcement learning algorithm where feature representations are learned by a neural network trained on a surrogate loss function (i.e. value), and a value function is learned on top of the feature representation using a "fast" least-squares algorithm. The authors prove the convergence of this method using methods from two time-scale stochastic approximation.

Convergent and stable nonlinear algorithms is an important problem in reinforcement learning, and this paper offers an interesting approach for addressing this issue. The idea of using a "fast" linear learner on top of a slowly changing representation is not new in RL (Levine et. al, 2017), but the authors somewhat motivate this approach by showing that it results in a stable and convergent algorithm. Thus, I view the convergence proof as the main contribution of the paper.

The paper is written clearly, but could benefit from more efficient use of space in the main paper. For example, I feel that the introduction and discussion in Section 3 on surrogate objectives could be considerably shortened, and a formal proof statement could be included from the appendix in Section 4, with the full proof in the appendix.

The experimental evaluation is detailed, and ablation tests show the value of different choices of surrogate loss for value function training, linear value function learning methods, and comparisons against other nonlinear algorithms such as DQN and Nonlinear GTD/TD/variants. A minor criticism is that it is difficult to position this work against the "simpler but not sound" deep RL methods, as the authors only compare to DQN on a non-standard benchmark task.

As additional related work, SBEED (Dai et. al, ICML 2018) also shows convergence for a nonlinear reinforcement learning algorithm (in the control setting), and quantifies the convergence rate while accounting for finite sample error. It would be good to include discussion of this work, although the proposed method and proofs are derived very differently.

---

> ### Author Response · Authors · 2018-11-24
> **Thank you for the review**
>
> Thank you for the helpful feedback.
>
> We agree that the TTN idea is straightforward---and likely already in use---but believe it is different from LS-DQN (Levine et al.). LS-DQN uses one head, and computes a fast linear update after a larger number of steps. The distinction is subtle, but important. The strategy from Levine et al. does not allow let us take advantage of the fast learning of linear methods (since it is only executed very infrequently) and further affects the learning of the features. In their paper, the FQI solution was only computed every 500000 steps (the DQN target net was updated every 10000) while TTN recomputes the weights every 10000 steps. The authors also mention that it was necessary to take only a small step in the direction of the recomputed weights so that it did not destabilize learning for the rest of the network. In TTNs, the fast linear weights do not affect the features, so we do not suffer from such stability problems.
>
> We have edited the paper to expand on the theory section and provided a formal theorem statement to clarify the contribution.
>
> We agree that a more thorough investigation of the control case would be beneficial in the future. Here, we are more concerned with the policy evaluation setting and provide some preliminary results for control to show the potential of the TTN approach.
>
> SBEED is a control algorithm which is shown to be stable with any differentiable function class (contrary to ours which is a prediction algorithm) using Nestrov’s smoothing technique (to overcome the structural limitations of the max Bellman operator) and a primal-dual formulation (to overcome the double sampling). However, it is important to note that the stability theorem provided in the SBEED paper claims only mean convergence. Ours is an almost sure convergence claim. We will however mention SBEED as related work.

---

> > ### Comment · AnonReviewer4 · 2018-11-25
> > **Response to rebuttal**
> >
> > Thank you for the clarifications. I do believe this work is important and relevant, and I have updated my score to recommend acceptance.

---

### Official Review · AnonReviewer1 · 2018-11-15
**Promising core idea**

**Rating:** 6
**Confidence:** 4

**Review:**

The paper introduces an algorithm (TTN) for non-linear online and on-policy value function approximation. The main novelty of the paper is to view non-linear value estimation as two separate components. One of representation learning from a non-linear mapping and one of linear value function estimation. The soundness of the approach stems from the rate at which each component is updated. The authors argue that if the non-linear component is updated at a slower rate than the linear component, the former can be viewed as fixed in the limit and what remains is a linear value function estimation problem for which several sound algorithms exist. TTN is evaluated on 4 domains and compared to several other value estimation methods as well as DQN on a control problem with two variations on the task's state space.

I'll start off the review by stating that I find the idea and theoretical justification of separating the non-linear and linear parts of value function estimation to be quite interesting, potentially impacting RL at large. Indeed, this view promises to reconcile latest developments in deep RL with the long-lasting work on RL with linear function approximators. However, there are a few unclear aspects that do not allow one to be fully convinced that this paper lives up to the aforementioned promise.

- For the theoretical contribution. The authors claim that the main challenge was to deal with the potentially dependent features outputted by the neural network. It is dealt with by using a projection that projects the linear parameters of the value function to a compact subset of the parameter space. Bar the appendix, there is no mention of this projection in the paper, on how this compact subset (that must include the optimal parameter) is defined and if this projection is merely a theoretical tool or if it was necessary to implement it in practice. There is a projection for the neural net weights too but I can see how for these it might not be necessary to use in practice. However, for the linear weights, as their computation potentially involves inverting ill-conditioned matrices, they can indeed blow-up relatively fast.

- I found the experimental validation to be quite rich but not done in a systematic enough manner. For instance, the experiment "utility of optimizing the MSPBE" demonstrates quite nicely the importance of each component but is only performed on a single task. As the theoretical analysis does not say anything about the improvements the representation learning can have on the linear value estimation nor if the loss used for learning the representation effectively yields better features for the MSPBE minimization, this experiment is rather important and should have been performed on more than a single domain.

Secondly, I do not find the chosen baselines to be sufficiently competitive. The authors state in Sec. 2 that nonlinear-GTD has not seen widespread use, but having this algorithm as the main competitor does not provide strong evidence that TTN will know a better fate. In the abstract, it is implied that outside of nonlinear-GTD, value function approximation methods are not sound. In approximate policy iteration algorithms such as DDPG or TRPO, there is a need in performing value estimation. It is done by essentially a fitted-Q iteration procedure which is sound. Why wasn't TTN compared to these methods? If it is because they are not online, why being online in the experiments of the paper is important? Showing that TTN is competitive with currently widespread methods for value estimated would have been more convincing than the comparison with nonlinear-GTD.

Thirdly, for the sake of reproducibility, as LSTD seems to be the method of choice for learning the linear part, it would have been adequate to provide an algorithm box for this version as is done for GTD2/TDC. LSTD is essentially a batch algorithm and there could be many ways to turn it into an online algorithm. With which algorithm were the results in the experimental section obtained?

Finally, on the control task, the authors add several modifications to their algorithm which results in an algorithm that is very close to that of Levine et al., 2017. Why was not the latter a baseline for this experiment? Especially since it was included in other experiments.

---

> ### Author Response · Authors · 2018-11-24
> **Thank you for the review**
>
> Thank you for the constructive feedback and comments.
>
> Concerning the projection step of the parameters, this is mainly a technical requirement for the proofs. This is not a strong requirement; we can initialize the compact subset arbitrarily and gradually increase it until it encompasses the whole parameter space (this is mentioned in remark 1 of appendix B). In practice, we do not utilize projection and simply let the parameters be unbounded.
>
> We have added results for the utility of optimizing the MSPBE for the other domains in the appendix.
>
> We have chosen to focus on evaluating theoretically-sound algorithms for policy evaluation. Many algorithms used for learning value functions in actor-critic methods either aggregate data from multiple agents to do nonlinear TD updates or use experience replay to resample minibatches with nonlinear TD updates. Nonlinear TD (and fitted Q-iteration for nonlinear Q) does not have any convergence guarantees ---even in the batch setting, so we did not include these variants. The convergence issues are due to the combination of nonlinear function approximation and bootstrapping the value targets, which is not solved by batching.
>
> We used the incremental version of LSTD which uses the Sherman-Morrison formula to do online updates on the A_inv matrix that LSTD requires (as is done in Section 3, page 10 of “Least squares policy evaluation algorithms with linear function approximation”, Nedic and Bertsekas 2002). We have added an algorithm box in appendix D to clarify this in the paper.
>
> For the control experiments, we have added results for the Levine et al. algorithm as another baseline in addition to vanilla DQN for nonimage catcher. Results for image catcher will also be added later.

---

### Meta-Review · Area_Chair1 · 2018-12-13
**Interesting work on approximation value function**

**Confidence:** 5
**Recommendation:** Accept (Poster)

**Metareview:**

The paper proposes a new method to approximate the nonlinear value function by estimating it as a sum of linear and nonlinear terms. The nonlinear term is updated much slower than the linear term, and the paper proposes to use a
fast least-square algorithm to update the linear term. Convergence results are also discussed and empirical evidence is provided.


As reviewers have pointed out, the novelty of the paper is limited, but the ideas are interesting and could be useful for the community. I strongly recommend taking reviewers comments into account for the camera ready and also add a discussion on the relationship with the existing work.

Overall, I think this paper is interesting and I recommend acceptance.